# ZFT is the major iron and zinc transporter in *Toxoplasma gondii*

**Dana Aghabi[1], Cecilia Gallego Rubio[1], Miguel Cortijo Martinez[1], Augustin Pouzache[1], Erin J Gibson[1,2], Lucas Pagura[1], Stephen J Fairweather[3], Giel G van Dooren[3], Clare R Harding[1]***

[1]Centre for Parasitology, School of Infection and Immunity, University of Glasgow, Glasgow, United Kingdom; [2]Wellcome-Wolfson Institute for Experimental Medicine, School of Medicine, Dentistry, and Biomedical Sciences, Queen's University Belfast, Belfast, United Kingdom; [3]Research School of Biology, Australian National University, Canberra, Australia

## eLife Assessment

This **important** study identifies a metal transporter in the plasma membrane of the obligate intracellular pathogen, *Toxoplasma gondii*. Using an array of different approaches, the authors **convincingly** demonstrate that this transporter mediates iron and zinc uptake and regulates diverse cellular processes, including parasite metabolism and differentiation. This work will be of broad interest to cell biologists and biochemists studying metal ion transport mechanisms.

***For correspondence:**
clare.harding@glasgow.ac.uk

**Competing interest:** The authors declare that no competing interests exist.

**Abstract** Transition metals, such as iron and zinc, are indispensable trace elements for eukaryotic life, acting as co-factors in essential processes ranging from metabolism to DNA replication. These metals can be transported into cells by an evolutionary-conserved family of metal transporters; however, how the ubiquitous mammalian parasite *Toxoplasma gondii* acquires essential metals has been unknown. Here, we have identified and characterised the first iron and zinc importer in *T. gondii*. This transporter, named ZFT, localised to the parasite plasma membrane and is essential for the parasite's life cycle. We find ZFT is regulated by iron availability and overexpression sensitises cells to excess iron and zinc. Using a conditional knockdown system, we find that knockdown of ZFT leads to reduction in mitochondrial respiration and a switch to a more quiescent lifecycle stage. To confirm transport activity, we find that knockdown of ZFT leads to a reduction in parasite-associated zinc and iron, and ZFT expression complements loss of zinc transporter activity in a yeast model. Further, expression of ZFT in *Xenopus* oocytes demonstrates direct uptake of iron, which is outcompeted in the presence of zinc. Overall, we have identified the first metal uptake transporter in *T. gondii* and demonstrated the importance of iron and zinc to the parasite. This finding advances our understanding of how this obligate intracellular parasite acquires nutrients from its host.

## Introduction

Iron, and other transition metals such as zinc, manganese, and copper, are essential nutrients for almost all life, playing vital roles in biological processes such as DNA replication, translation, and metabolic processes including mitochondrial respiration (*Teh et al., 2024*). As metals cannot diffuse across membranes, organisms have developed a range of strategies to take up metals, for example through endocytosis of iron-bound transferrin by mammalian cells, or siderophore uptake systems in bacteria. However, iron and other transition metals can catalyse reactive oxygen species, and so intracellular levels are tightly regulated (*Galaris et al., 2019*; *Imlay et al., 1988*).

*Toxoplasma gondii* is an obligate intracellular apicomplexan parasite which chronically infects around 30% of the world's population (*Bisetegn et al., 2023*). Although typically asymptomatic, toxoplasmosis can be dangerous in immunocompromised individuals and to the developing foetus, resulting in severe congenital defects (*Desmonts and Couvreur, 1974*; *Wong and Remington, 1994*). Unusually among the Apicomplexa, *T. gondii* is promiscuous, capable of infecting all nucleated cells (*Tenter et al., 2000*) and all warm-blooded species, causing significant economic losses in sheep and goat farming (*Stelzer et al., 2019*). Iron is essential for the parasite where it is used in essential cellular processes, including haem biosynthesis (*Bergmann et al., 2020*; *Kloehn et al., 2021*) and iron–sulphur (Fe–S) cluster biogenesis pathways (*Aw et al., 2021*) and depletion of iron using chelators restricts parasite replication in vitro and in vivo (*Aghabi et al., 2023*; *Ferrer et al., 2012*; *Mahmoud, 1999*; *Raventos-Suarez et al., 1982*; *Sloan et al., 2025*). Beyond iron, the role of other transition metals in *Toxoplasma* biology is less well understood. It is predicted that at least 800 *Toxoplasma* proteins require zinc, several of which have already been shown to be essential (*Gissot et al., 2017*; *Semenovskaya et al., 2020*) and loss of zinc storage leads to mild parasite defects (*Chasen et al., 2019*). However, the transport pathways, use of zinc and effects of zinc depletion have not been investigated. Further, although both iron and zinc are essential for *Toxoplasma*, the mechanisms of their uptake in *T. gondii* have not yet been established.

The ZIP (Zrt-/Irt-like protein)-domain containing proteins represent a large, highly conserved family of metal transporters (*Guerinot, 2000*). ZIP-domain proteins mediate both iron and zinc uptake and intracellular trafficking in mammalian, plant and fungal cells. ZIP-domain containing proteins have also been utilised in eukaryotic parasites; a *Leishmania* ZIP-domain containing protein was identified as the major iron importer and was essential for intracellular replication (*Huynh et al., 2006*; *Jacques et al., 2010*). In *Plasmodium,* the plasma membrane-localised ZIP-domain containing protein ZIPCO was shown to be essential only in the liver stage of the parasite and was identified as a potential iron and zinc transporter based on partial rescue of a growth phenotype with iron and zinc (*Sahu et al., 2014*). In the blood stages, iron appears to be transported from the *Plasmodium* food vacuole to the cytosol by DMT1, where it could be utilised by the apicoplast and mitochondrion (*Loveridge and Sigala, 2024*; *Zhong and Zhou, 2023*).

*T. gondii* encodes four predicted ZIP-domain containing proteins: TGME49_261720, TGME49_254080, TGME49_500163, and TGME49_225530, none of which have been previously characterised. Our previous work identified that TGME49_261720 mRNA was stabilised under low iron by the presence of a stem-loop structure in the 3' UTR (*Sloan et al., 2025*). Loss of this regulation inhibited parasite survival under low iron; however, the function of the protein was not investigated. Here, we characterise TGME49_261720, named **Z**n and **F**e **T**ransporter (ZFT), in detail. We find ZFT localisation and abundance is dynamic and dependent on parasite replication, vacuolar stage and iron availability. ZFT is essential for the parasite's lytic lifecycle and knockdown leads to inhibition of mitochondrial respiration, defects in apicoplast biogenesis and initiation of stage conversion. Unlike in *Plasmodium*, ZFT knockdown cannot be rescued by addition of exogenous iron or zinc. By quantifying parasite-associated metals, we show ZFT knockdown parasites have significantly less iron and zinc. Confirming transport activity, we find expression of ZFT rescues zinc uptake in mutant yeast and transports iron upon expression in *Xenopus*. These data strongly support the role of ZFT as the major iron and zinc transporter in *T. gondii*.

## Results

### Iron transport in *T. gondii*

We examined the genome of *Toxoplasma* and identified four genes encoding predicted ZIP-domains (*Amos et al., 2022*). Of these, both TGME49_254080 and TGME49_500163 were predicted to contain a non-standard number of transmembrane domains (6 and 3, respectively, rather than 8 or 9) and the ZIP-domain of TGME49_254080 was relatively poorly conserved and was not considered further. Both TGME49_261720 and TGME49_225530 encoded ZIP-domains with high predicted confidence ($E$ value $<7 \times 10^{-30}$) are predicted to have eight transmembrane domains and share limited sequence similarity with PfZIPCO (21% and 26% identity, respectively). We examined the sequence conservation at the binuclear metal centres (BMC) M1 and M2 in transmembrane domains α4 and α5, which have been shown to be required for metal binding in a bacterial ZIP homolog (BpZIP) (*Wiuf et al.,*

*2022*). At M1, *Toxoplasma* TGME49_261720 contains an unusual HGxxEG motif, which we found only in coccidian ZIPs and in some marine organisms (e.g. *Branchiostoma floridae*), the impact of which is not clear (*Figure 1a*). In contrast, TGME49_225530 contains a HAxxEG motif, also found in prokaryotic ZIP9-like proteins (*Hu, 2021*). Interestingly, the apicomplexan ZIP-domain proteins examined have a HK motif at the M2 metal-binding site (*Figure 1a*), including those from *Cryptosporidium* and the digenic gregarine *Porospora*; however, this is not conserved in the free-living alveolata such as *Chromera*. The HK M2 motif is characteristic of the ZIPI family of ZIP-domain proteins, shared in HsZIP1 and HsZIP2 (*Dufner-Beattie et al., 2003*). This lysine likely prevents metal binding at this site; however, it has been shown to be required for transport activity in HsZIP2, possibly with a role in pH responsivity of the transporter (*Gyimesi et al., 2019*). Based on the unusual conserved motif, the predicted plasma membrane localisation, essentiality in tachyzoites and our previous results demonstrating iron-mediated regulation (*Sloan et al., 2025*), we decided to functionally characterise the role of TGME49_261720. We constructed a structural prediction of TGME49_261720 as a dimer (in common with other reported ZIP-domain containing proteins). TGME49_261720 had 8 transmembrane domains with a large, unstructured loop between helices 3 and 4, predicted to be on the outer lumen, the role of which is unclear. The highly conserved metal binding residues H314 and E318 are indicated within the core of the predicted sequence (*Figure 1b*).

## ZFT localisation is dynamic and vacuolar stage dependent

We have previously shown that the 3′ UTR of ZFT plays an important role in iron-mediated regulation (*Sloan et al., 2025*) and so we tagged ZFT with a 3xHA epitope tag with the endogenous 3′ UTR (ZFT-3HA$_{zft}$) as previously described (*Sloan et al., 2025*; *Figure 1c*). Western blot demonstrated the protein ran at the expected size of around 50 kDa, although with multiple, non-specific lower bands (asterisks) (*Figure 1d*). Using immunofluorescence, we found that ZFT localised to the periphery of the parasite with a second foci at the basal pole of the cell (*Figure 1e*). To determine if changing the UTR affected ZFT localisation, we also tagged it with a 3xHA epitope tag followed by the *sag1* 3′ UTR (ZFT-3HA$_{sag1}$) as previously described (*Sloan et al., 2025*) and saw a similar localisation (*Figure 1—figure supplement 1a, b*). Interestingly, localisation was highly dependent on the number of parasites per vacuole. In the early stages (2–4 parasites/vacuole), ZFT localised peripherally; however, this shifted to basal localisation in larger vacuoles (>8 parasites) (*Figure 1e, f*). Accompanying this change in localisation, we quantified levels of ZFT-3HA$_{zft}$ and found the mean fluorescence intensity (MFI) of ZFT/parasite was significantly ($p < 0.001$, one-way ANOVA, Tukey corrected for multiple comparisons) higher in vacuoles with fewer parasites, compared to larger vacuoles (eight parasites) (*Figure 1g*). This suggests that ZFT is degraded in a vacuolar stage-dependent manner. Previously, we showed that ZFT mRNA levels are regulated by low iron (*Sloan et al., 2025*). To determine if the localisation of ZFT was affected under iron starvation, we treated the parasites with 100 mM of the iron chelator deferoxamine (DFO) for 24 hr. Upon iron starvation, ZFT-3HA$_{zft}$ localisation was maintained at the parasite periphery, although some intracellular vesicles could be observed (*Figure 1—figure supplement 1c*). Under these conditions, we do not see parasite replication and so to confirm this, we used a lower dose of DFO (20 mM) which allowed for limited parasite replication (*Hanna et al., 2025*). Again, we saw significantly increased peripheral localisation ($p = 0.0021$, two-way ANOVA, Fisher LSD) under low iron (*Figure 1—figure supplement 1d, e*), supporting our hypothesis that ZFT localisation is both vacuole stage- and iron-dependent.

These data show ZFT expression and localisation is dynamic depending on vacuolar stage.

## ZFT expression is regulated by excess iron

The localisation and expression of transporters is often regulated by substrate availability (*Conte and Walker, 2011*; *Gao and Dubos, 2021*; *Lee et al., 2020*), and we have previously shown the iron transporter VIT is regulated by iron in *Toxoplasma* (*Aghabi et al., 2023*). We have previously demonstrated the *zft* 3′ UTR is sufficient to stabilise the mRNA under low iron (*Sloan et al., 2025*); however, expression under excess iron was not examined. To determine if ZFT-3HA$_{zft}$ is regulated by excess iron, we treated parasites with exogenous ferric ammonium chloride (FAC). At 18 hr post treatment, we quantified the major ZFT-3HA$_{zft}$ band and found that levels were slightly (to around 87% of the untreated) but significantly ($p = 0.028$, two-tailed paired *t* test) reduced (*Figure 2a, b*). However, by 24 hr post treatment, we found that ZFT-3HA$_{zft}$ expression was reduced to around 40%

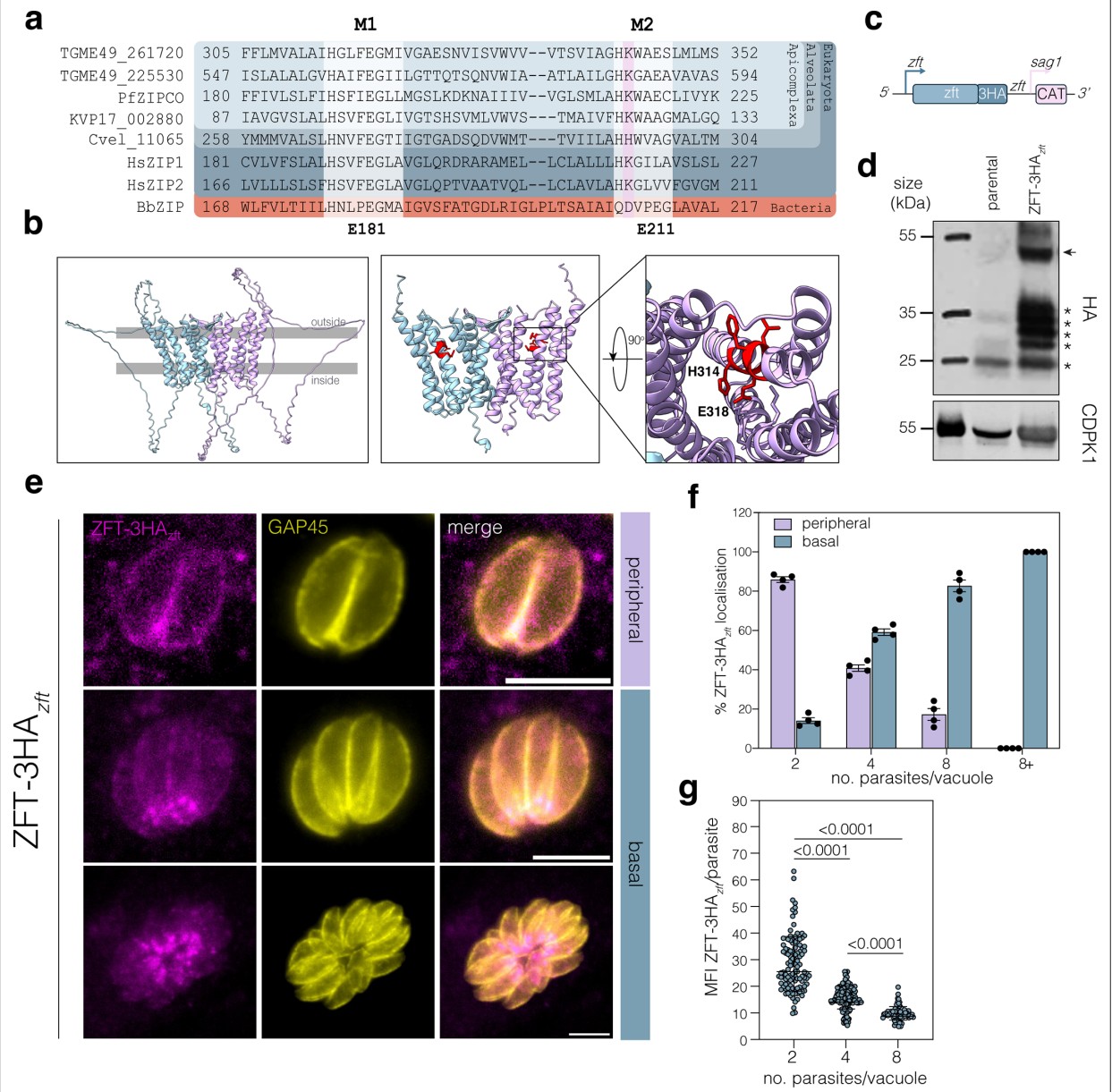

**Figure 1.** ZFT is a ZIP-domain containing protein with dynamic localisation. (**a**) Alignment of ZIP proteins from *T. gondii* (TGME49_261720), *T. gondii* (TGME49_266800), *P. falciparum* (PF3D7_1022300), *Porospora* cf. gigantea (KVP17_002880), *Chromera vella* (Cvel_11065), *H. sapiens* (HsZIP1 and HsZIP2), and *Bordetella bronchiseptica* (BpZIP). Key conserved residues in the binuclear metal centres (BMC) M1 and M2 in transmembrane domains a4 and a5, which have been shown to be required for metal binding, are highlighted. HK motif found in Apicomplexa and HsZIP1 and HsZIP2 highlighted. Conserved glutamate residues highlighted listed below, numbers from BbZIP sequence. (**b**) Alphafold model of TGME49_261720 dimer with key residues highlighted (**c**) Schematic of F3ΔHX ZFT-3HA_zft strain construction to endogenously tag ZFT with 3xHA epitope tags at the C-terminal under the endogenous promoter with the endogenous 3' UTR. (**d**) Western blot analysis of ZFT-3HA_zft confirming the expected band size (50 kDa). Non-specific bands indicated with asterisks. CDPK1 (TGME49_301440) used as a loading control. (**e**) Immunofluorescence of ZFT-3HA_zft demonstrating dynamic peripheral or basal localisation. Scale bars 5 µm. (**f**) Percentage of ZFT-3HA_zft localisation (peripheral or basal) in respect to number of parasites per vacuole. 100 parasites were counted for each condition. Bars at mean of four independent experiments, ± SD. (**g**) Mean fluorescence intensity (MFI) of ZFT-3HA_zft per parasite in relation to the number of parasites per vacuoles. Each point represents individual MFI/parasite values from three independent experiments. Bars at mean of the experiments ± SD. p values from ordinary one-way ANOVA, Tukey corrected for multiple comparisons.

The online version of this article includes the following source data and figure supplement(s) for figure 1:

**Source data 1.** PDF file containing original western blots for *Figure 1D* indicating the relevant bands and conditions.

**Source data 2.** Original files for western blot analysis displayed in *Figure 1D*.

**Figure supplement 1.** Localisation of ZFT-HA is not dependent on the 3' UTR, but is dependent on iron availability.

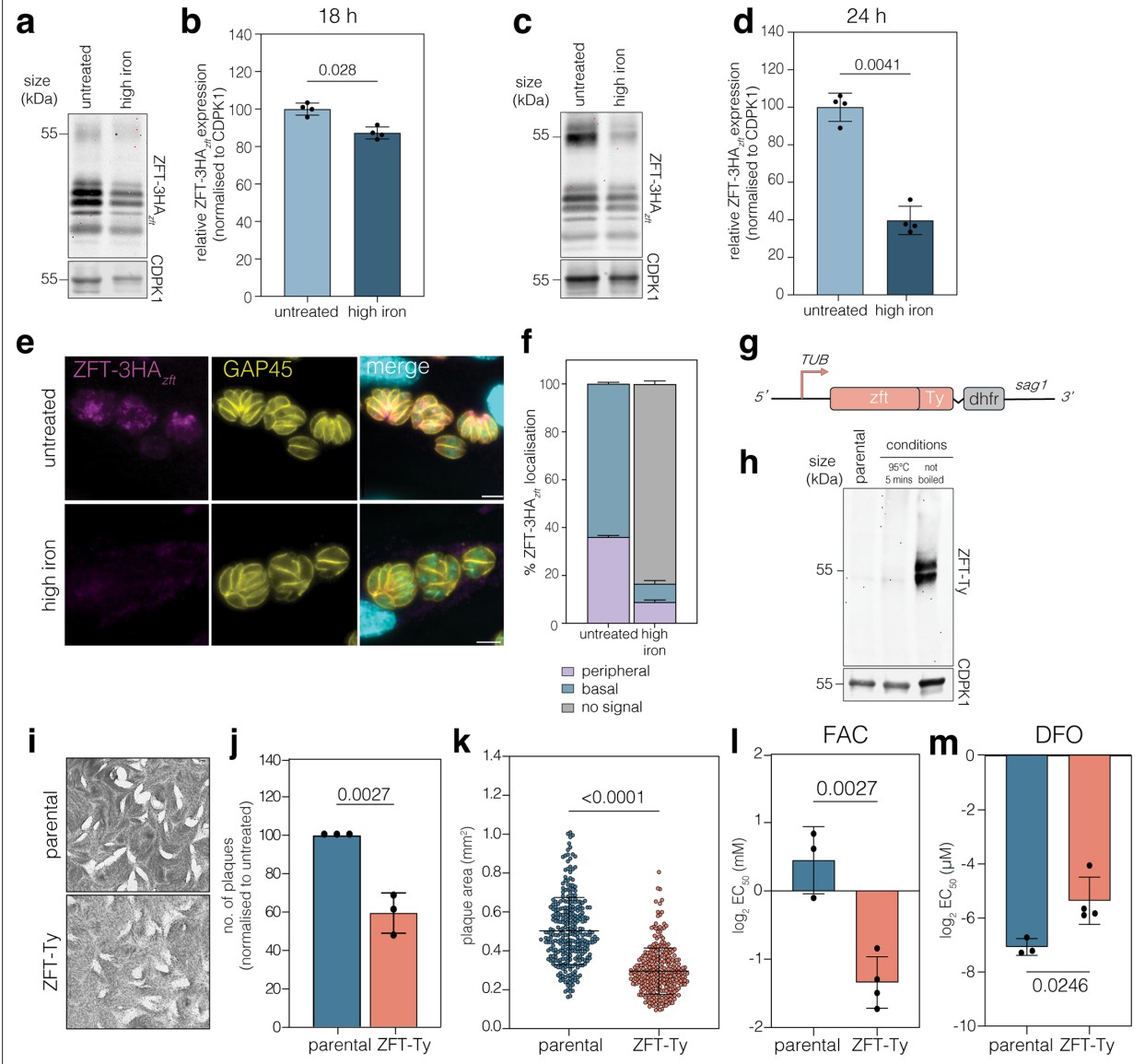

**Figure 2.** ZFT-3HA$_{zft}$ is regulated by iron, and overexpressing ZFT leads to a change in iron sensitivity. (**a**) Western blot analysis showing ZFT-3HA$_{zft}$ expression levels 18 hr post infection in untreated and high iron (500 µM FAC) conditions. CDPK1 was used as a loading control. (**b**) Quantification of ZFT-3HA$_{zft}$ from three independent experiments. Bars at mean ± SD. p values from two-tailed paired $t$ test. (**c**) As above, at 24 hr post infection in untreated and high iron (500 µM FAC) conditions. (**d**) Quantification of ZFT-3HA$_{zft}$ from three independent experiments. Bars at mean ± SD. p values from two-tailed paired $t$ test. (**e**) Immunofluorescence assay of ZFT-3HA$_{zft}$ at 24 hr post infection in untreated and high iron (500 µM FAC) conditions. Scale bars 5 µm. (**f**) Quantification of ZFT-3HA$_{zft}$ localisation (peripheral/basal or no signal) in untreated and high iron conditions. Bars at mean of three independent experiments, ± SD. (**g**) Schematic of ZFT-Ty strain construction to overexpress ZFT from the TUB8 promoter in RHtdTomato parental line. (**h**) Western blot of ZFT-Ty showing expected size. (**i**) Plaque assay of parental and ZFT-Ty parasites in untreated conditions. Quantification of plaque number (**j**) and area (**k**) for parental and ZFT-Ty parasites, p values are unpaired two-tailed $t$ test. Points represent individual plaque areas from three independent experiments, bars at mean ± SD. p values from two-tailed unpaired $t$ test with Welch's correction. Graphs showing mean parasite EC$_{50}$ for FAC (**l**) and DFO (**m**) for RHtdTomato and ZFT-Ty parasites. Each point represents an independent experiment, bars at mean of $n$ = 3, ± SD. p values from two-tailed unpaired $t$ test.

The online version of this article includes the following source data and figure supplement(s) for figure 2:

**Source data 1.** PDF file containing original western blots for *Figure 2A, C, H* indicating the relevant bands and conditions.

**Source data 2.** Original files for western blot analysis displayed in *Figure 2A, C, H*.

**Figure supplement 1.** Characterisation of ZFT-Ty overexpression line.

of the untreated (p = 0.0041, two-tailed paired $t$ test) (*Figure 2c, d*). We confirmed loss of ZFT-3HA$_{zft}$ expression under excess iron by immunofluorescence (*Figure 2e, f*), demonstrating ZFT expression is significantly reduced under excess iron.

## Overexpression of ZFT leads to a change in iron sensitivity

Initial attempts to overexpress ZFT were unsuccessful, suggesting that overexpression was toxic. However, we were able to stably overexpress ZFT by coupling a second copy of ZFT-Ty expressed from a strong promoter, pTUB8 (*Frénal et al., 2010*) with DHFR using a P2A-slip peptide (*Markus et al., 2019*) and maintaining the line in pyrimethamine (*Figure 2g*). We validated the presence of the second copy by PCR (*Figure 2—figure supplement 1a*) and the protein by western blotting (*Figure 2h*). Although ZFT-Ty was detected at the correct size, the protein was undetectable in boiled samples. This may suggest that ZFT precipitates out of solution when overexpressed. Indirect immunofluorescence assay (IFA) demonstrated that ZFT-Ty localised to the periphery and intracellular structures (*Figure 2—figure supplement 1b*). To determine the effect of overexpression of ZFT on parasite fitness, we performed a plaque assay of both parental and ZFT-Ty parasites (*Figure 2i*). Parasites overexpressing ZFT form significantly (p = 0.0027, unpaired two-tailed $t$ test) fewer plaques than parental parasites (*Figure 2j*), with significantly (p < 0.0001, two-tailed unpaired $t$ test with Welch's correction) smaller area (*Figure 2k*), confirming that overexpression of ZFT inhibits parasite replication. Overexpression of metal transporters can alter the sensitivity to environmental metal conditions (*Connolly et al., 2002*; *Li and Kaplan, 1998*). To test this, we quantified the effects of ZFT overexpression on iron sensitivity. Our data show parasites overexpressing ZFT-Ty are significantly (p = 0.0027, two-tailed unpaired $t$ test) more sensitive (EC$_{50}$ = 0.386 mM, 95% C.I. 0.315–0.479 mM) to excess iron, compared to the parental line (EC$_{50}$ = 1.29 mM, 95% C.I. 0.813–2.01 mM) (*Figure 2l*, *Figure 2—figure supplement 1c*). Excess iron leads to parasite death largely through oxidative stress (*Aghabi et al., 2023*). To confirm that parasites were not more sensitive based on non-specific increased sensitivity to oxidative stress, we treated parasites with sodium arsenite (Ars), which has previously been used to induce oxidative stress in *T. gondii* (*Aghabi et al., 2023*; *Augusto et al., 2021*). However, overexpression of ZFT does not lead to any significant change in oxidative stress sensitivity (*Figure 2—figure supplement 1d, e*), suggesting the phenotype is specific to iron. We then tested how ZFT-Ty overexpression affected sensitivity to iron starvation using DFO. We found parasites overexpressing ZFT are significantly (p = 0.0246, two-tailed unpaired $t$ test) more resistant (EC$_{50}$ = 22.81 μM, 95% C.I. 16.8–30.6 μM) to iron chelation, compared to the parental line (EC$_{50}$ = 8.18 μM, 95% C.I. 6.94–9.71 μM) (*Figure 2m*, *Figure 2—figure supplement 1f*). These data suggest that overexpression of ZFT leads to alterations in parasite iron sensitivity.

## ZFT is essential for parasite survival

To determine the importance of ZFT in parasite replication, an inducible knockdown line was generated by replacing the promoter of ZFT with the T7S4 promoter which is inhibited in the presence of ATc (*Sheiner et al., 2011*; *Figure 3a*), which was validated by PCR (*Figure 3—figure supplement 1a*). To verify conditional knockdown, we performed RT-qPCR as previously described (*Sloan et al., 2025*). By 24 hr post induction with ATc, we found that *zft* mRNA was significantly (p = 0.0032, two-tailed unpaired $t$ test with Welch's correction) reduced (*Figure 3b*). We epitope-tagged ZFT in the T7S4-ZFT background (*Figure 3c*) and confirmed integration by PCR (*Figure 3—figure supplement 1b*). We found that ZFT-3HA$_{sag1}$ was undetectable by immunofluorescence by 48 hr post induction (*Figure 3d*). Western blotting demonstrated that swap of the promoter led to a significant (around five times) overexpression of ZFT in the T7S4-ZFT cell line, compared with the endogenous promoter (*Figure 3—figure supplement 1c, d*). We validated the ATc-induced inhibition of expression by western blot (*Figure 3e*) and show that there is a significant (p < 0.0001, two-tailed paired $t$ test) drop in ZFT-3HA$_{sag1}$ expression levels by 24 hr post induction. At 48 hr after adding ATc, only around 10% of the protein remained (p = 0.0131, two-tailed paired $t$ test) (*Figure 3f*). Previous studies on both bacterial and eukaryotic ZIPs suggest that these proteins function in either homodimers or heterodimers (*Ahern et al., 2019*; *Bin et al., 2011*; *Lin et al., 2010*; *Taylor et al., 2016*). To determine if ZFT-3HA$_{sag1}$ forms a complex in *Toxoplasma*, we performed a Blue Native PAGE (BN-PAGE) and found that ZFT-3HA$_{sag1}$ forms a complex of ~100 kDa, which is lost upon knockdown, suggesting the formation of a dimer (*Figure 3g*, uncropped in *Figure 3—figure supplement 1e*).

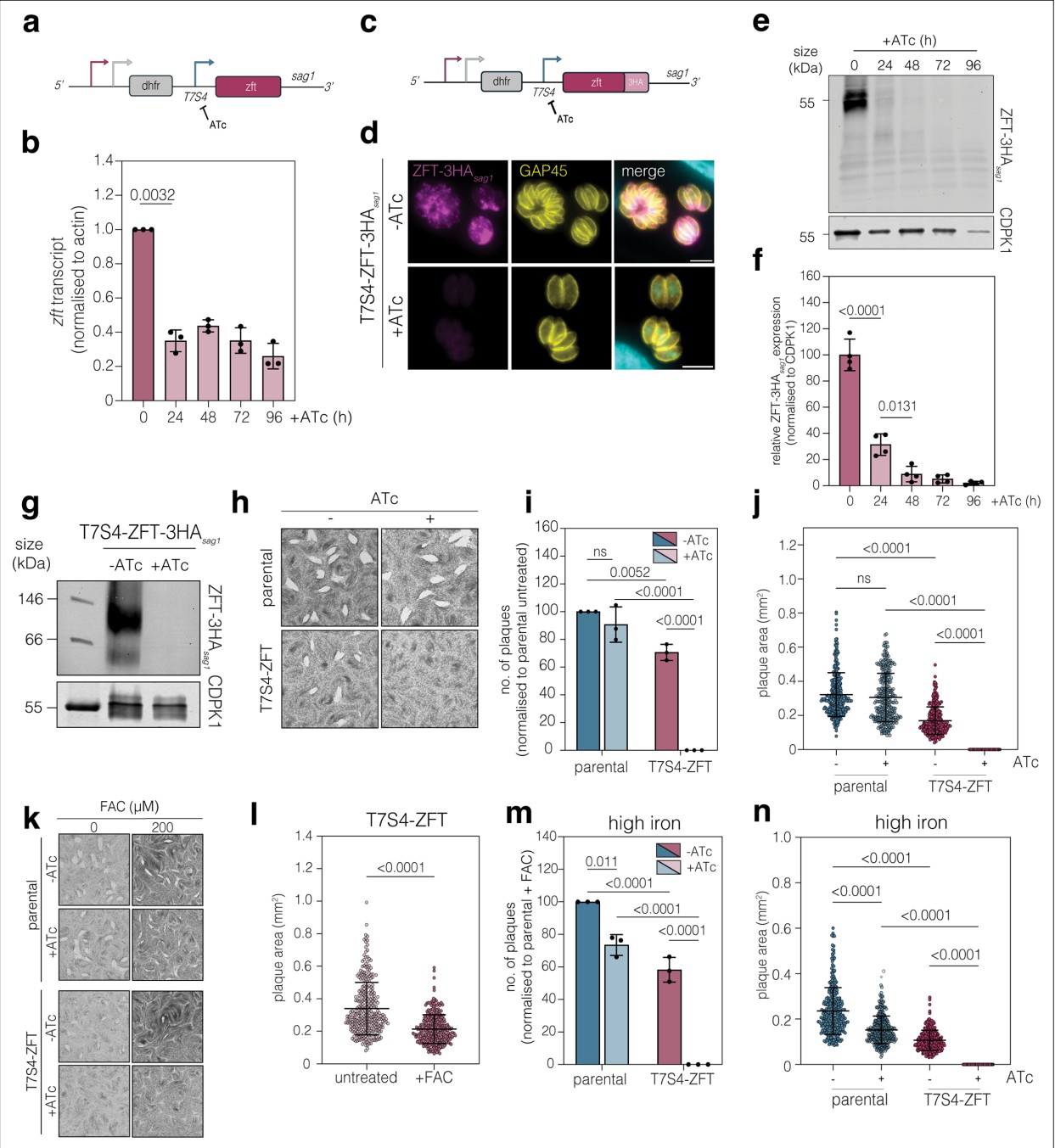

**Figure 3.** ZFT knockdown blocks parasite replication. (**a**) Schematic of F3ΔHX T7S4-ZFT under the T7S4 inducible promoter. (**b**) RT-qPCR showing ZFT mRNA abundance, relative to 0 hr, normalised to actin. Bars at mean of three independent experiments, ± SD. p values from two-tailed unpaired *t* test. (**c**) Schematic of F3ΔHX T7S4-ZFT-3HA$_{sag1}$ under the T7S4 inducible promoter. (**d**) Immunofluorescence assay showing T7S4-ZFT-3HA$_{sag1}$ expression in untreated parasites and after induction with ATc for 48 hr. Scale bar 5 μm. (**e**) Western blot demonstrating successful knockdown of ZFT at 24, 48, 72 and 96 hr post ATc induction. CDPK1 used as a loading control. (**f**) Quantification of (**e**), relative to 0 hr, normalised to CDPK1. Bars at mean of four independent experiments, ± SD. p values from two-way ANOVA, Tukey corrected for multiple comparisons. (**g**) Blue Native PAGE of T7S4-ZFT-3HA$_{sag1}$, untreated and +ATc 48 hr induction, showing a ZFT-HA complex at around 100 kDa. CDPK1 used as a loading control. (**h**) Plaque assay of parental and T7S4-ZFT parasites untreated and ATc treated, 6 days, showing knockdown of ZFT renders the parasite unable to form plaques in the host cell monolayer. (**i**) Quantification of (**h**) showing number of plaques, normalised to parental untreated. Bars at mean of three independent experiments, ± SD. p values from two-way ANOVA, Tukey corrected for multiple comparisons. (**j**) Quantification of plaques area from (**h**). Points represent individual plaques from three independent experiments. Bars at mean ± SD. p values from ordinary one-way ANOVA, Tukey corrected for multiple comparisons. (**k**) Plaque assays with addition of 200 μM FAC do not rescue ZFT knockdown parasites. (**l**) Quantification of plaque area generated by T7S4-ZFT

*Figure 3 continued on next page*

*Figure 3 continued*

parasites in untreated and high iron (200 µM FAC) conditions. Points represent individual plaque areas from three independent experiments. Bars at mean ± SD. p values are from two-tailed unpaired *t* test with Welch's correction. (**m**) Quantification of plaque number, normalised to parental untreated. Bars at mean of three independent experiments, ± SD. p values from two-way ANOVA, Tukey corrected for multiple comparisons. (**n**) Quantification of plaque area generated by parental and T7S4-ZFT parasites with and without ATc, in the presence of excess iron (200 µM FAC). Points represent individual plaque areas from three independent experiments. Bars at mean ± SD. p values from ordinary one-way ANOVA, Tukey corrected for multiple comparisons.

The online version of this article includes the following source data and figure supplement(s) for figure 3:

**Source data 1.** PDF file containing original western blots for *Figure 3E, G* indicating the relevant bands and conditions.

**Source data 2.** Original files for western blot analysis displayed in *Figure 3E, G*.

**Figure supplement 1.** Creation and validation of the ZFT conditional knockdown line.

**Figure supplement 1—source data 1.** PDF file containing original western blots for *Figure 3—figure supplement 1c and e* indicating the relevant bands and conditions.

**Figure supplement 1—source data 2.** Original files for western blot analysis displayed in *Figure 3—figure supplement 1c and e*.

To examine the role of ZFT, a plaque assay was performed (*Figure 3h*). Replacement of the ZFT promoter alone led to a significant decrease in plaque number and area (p = 0.0052 and p < 0.0001, respectively, two-way ANOVA, Tukey corrected for multiple comparisons) (*Figure 3h, i and j*). This, combined with our previous study (*Sloan et al., 2025*), demonstrates that correct expression of ZFT is important for parasite fitness. Addition of ATc renders T7S4-ZFT parasites unable to form plaques in the host cell monolayer (*Figure 3h, i, j*), confirming that expression of ZFT is essential for the parasite. In *Plasmodium*, the growth defect upon loss of DMT1 (*Loveridge and Sigala, 2024*) or ZIPCO (*Sahu et al., 2014*) was able to be rescued by addition of exogenous substrate. To determine if loss of ZFT could be rescued by addition of iron, we treated ZFT knockdown cells with 200 µM FAC and quantified growth (*Figure 3k*). We found that replacing the ZFT promoter alone led to a significant reduction in plaque area (p < 0.0001, two-tailed unpaired *t* test with Welch's correction) in high iron conditions (200 µM FAC) compared to untreated conditions (*Figure 3l*), potentially due to the overexpression of ZFT under the T7S4 promoter. However, exogenous iron was unable to rescue the growth defect, either by plaque number or area (*Figure 3k, m, n*). To determine if iron could restore growth when some ZFT remained, we also performed a plaque assay with half the standard concentration of ATc (0.25 µM) (*Figure 3—figure supplement 1f*). We find that in these conditions, ZFT KD parasites are still unable to form any plaques, even in the presence of excess iron (200 µM FAC) (*Figure 3—figure supplement 1f–h*). This suggests a strong, irreversible growth defect and the essentiality of ZFT as the major iron transporter.

## ZFT knockdown alters organellar function

Iron is required for proteins essential in parasitic organelles such as the apicoplast and mitochondrion (*Maclean et al., 2024*; *Pamukcu et al., 2021*; *Renaud et al., 2022*) and disruption of organellar function can lead to morphological changes (*Mallo et al., 2021*; *Wu et al., 2022*). To determine if ZFT knockdown was associated with changes in the apicoplast, we initially imaged at 24 and 48 hr of ATc; however, we saw no clear morphological defects (*Figure 4a*). To determine if there is a defect in apicoplast protein import post-ZFT depletion, we examined CPN60 processing at 48 hr treatment and saw no change (*Figure 4—figure supplement 1a*). We also examined protein lipoylation, as the apicoplast contains the essential Fe–S–protein LipA, which is required for lipoylation of the apicoplast-localised pyruvate dehydrogenase E2 (PDH-E2) (*Pamukcu et al., 2021*; *Crawford et al., 2006*). Using an anti-lipoic acid antibody, we were able to detect lipoylated PDH-E2 at 48 hr post ATc treatment (*Figure 4b*) and saw no significant change in PDH-E2 lipoylation (*Figure 4—figure supplement 1b*), suggesting that at this time point, apicoplast functions are maintained. We also detected the two mitochondrially localised lipoylated enzymes, branched-chain 2-oxo acid dehydrogenase-E2 (BDCH-E2) and 2-oxoglutarate dehydrogenase-E2 (OGDH-E2) (*Figure 4b*). Mitochondrial lipoylation occurs through iron-independent lipoic acid scavenging (*Crawford et al., 2006*) and although lipoylation of OGDH-E2 does not appear to significantly change (*Figure 4—figure supplement 1c*), interestingly lipoylated BDCH-E2 is significantly (p = 0.027, two tailed unpaired *t* test) increased upon ZFT depletion (*Figure 4—figure supplement 1d*). Given the previously established delayed-death phenotype

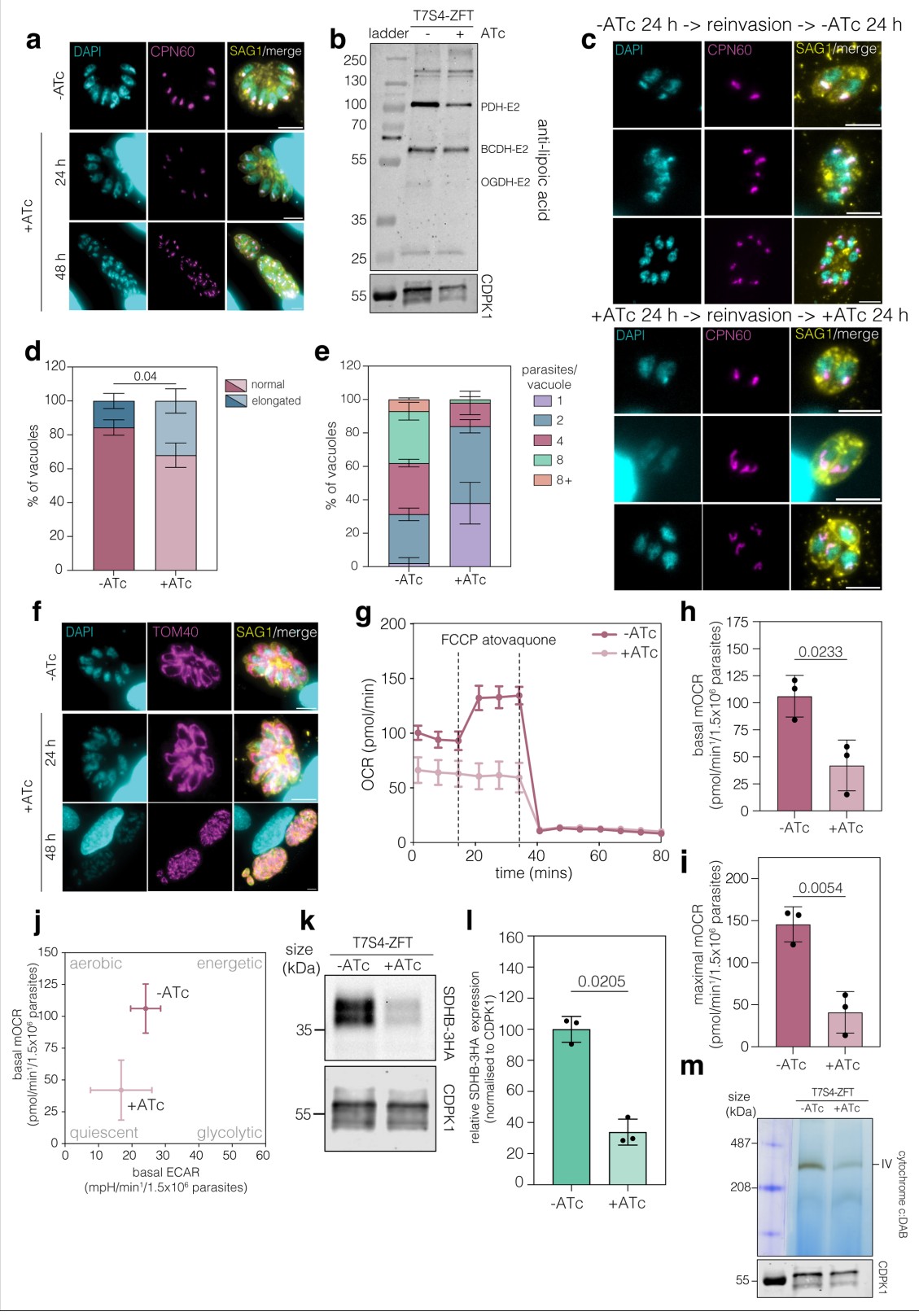

**Figure 4.** ZFT knockdown reduces mitochondrial respiration and the expression of Fe–S protein SDHB. (**a**) Immunofluorescence assay of T7S4-ZFT-3HA*sag1* in the absence and presence of ATc for 24 and 48 hr, staining for CPN60 (apicoplast marker). (**b**) Western blot analysis of T7S4-ZFT with and without ATc 48 hr staining for lipoic acid. TgPDH-E2, TgBCDH-E2 and TgOGDH-E2 were detected 48 hr post ZFT knockdown, suggesting the apicoplast is functional. (**c**) Immunofluorescence assay of T7S4-ZFT parasites with and without ATc staining for CPN60 (apicoplast marker) following a further

*Figure 4 continued on next page*

*Figure 4 continued*

round of mechanical release and invasion to examine if ZFT knockdown has a delayed apicoplast phenotype. Scale bar 5 µm. (**d**) Quantification of % of apicoplasts from (**c**) with and without ATc that appeared normal or elongated. 100 vacuoles for each condition were quantified. Bars at mean of three independent replicates, ± SD. (**e**) Quantification of the percentage of parasites per vacuole with and without ATc following a further round of mechanical release and invasion. 100 vacuoles for each condition were counted. Bars at mean of three independent replicates, ± SD. (**f**) Immunofluorescence assay as above, staining for TOM40 (mitochondrial marker). Scale bar 5 µm (**g**) Mitochondrial oxygen consumption rate (mOCR) of T7S4-ZFT, untreated or ATc treated for 48 hr via Seahorse assay. Quantification of basal mOCR (**h**) and maximal mOCR (**i**). Bars at mean of three independent experiments, ± SD. p values are from two-tailed unpaired *t* test with Welch's correction. (**j**) Metabolic map demonstrating ZFT knockdown shifts parasites to a more quiescent state. Points at mean, ± SD. (**k**) Western blot of T7S4-ZFT SDHB-3HA, untreated and ATc treated for 48 hr. CDPK1 used as a loading control. (**l**) Quantification of SDHB-3HA expression, relative to uninduced, normalised to CDPK1. Bars at mean of three independent experiments, ± SD. p values are two-tailed paired *t* test. (**m**) Complex IV activity assay of T7S4-ZFT parasites with and without ATc 48 hr showing ZFT knockdown reduces complex IV activity. CDPK1 used as a loading control.

The online version of this article includes the following source data and figure supplement(s) for figure 4:

**Source data 1.** PDF file containing original western blots for *Figure 4B, K, M* indicating the relevant bands and conditions.

**Source data 2.** Original files for western blot analysis displayed in *Figure 4B, K, M*.

**Figure supplement 1.** Validation of lines for testing mitochondrial and apicoplast functions in ZFT knockdown.

**Figure supplement 1—source data 1.** PDF file containing original western blots for *Figure 4—figure supplement 1a and j* indicating the relevant bands and conditions.

**Figure supplement 1—source data 2.** Original files for western blot analysis displayed in *Figure 4—figure supplement 1a and j*.

of the apicoplast, we determined the effects of ZFT knockdown on apicoplast maintenance after a further round of mechanical release and invasion. Here, we observed a small but significant increase (p = 0.04, two-way ANOVA, Tukey corrected for multiple comparisons) in elongated apicoplasts (*Figure 4c, d*). We also quantified parasite replication and found a significant (p < 0.0055 at the 1, 2, 4, and 8 parasites/vacuole, two-way ANOVA with Šidák correction) decrease in parasites/vacuole in the second round of invasion, compared to untreated (*Figure 4e*).

Iron is also required within the parasite mitochondrion (*Maclean et al., 2024*; *Pamukcu et al., 2021*). We first examined gross parasite mitochondrial morphology by light microscopy (*Figure 4f*) and saw no significant alterations. To assess if ZFT knockdown affects mitochondrial respiration, oxygen consumption rate (OCR) was quantified using a Seahorse assay (*Hayward et al., 2022*; *Figure 4g*). We found that 48 hr of ZFT knockdown significantly reduced basal (*Figure 4h*) and maximal (*Figure 4i*) mitochondrial oxygen consumption rates (mOCRs) (p = 0.0233, p = 0.0054, respectively, two-tailed unpaired *t* test with Welch's correction), compared to untreated parasites. Interestingly, we find ZFT knockdown does not significantly change the extracellular acidification rate (ECAR), a proxy for glycolysis (*Hayward et al., 2022*; *Figure 4—figure supplement 1e*) and the metabolic map suggests that ZFT knockdown leads to a reduction in oxidative phosphorylation and a more quiescent phenotype (*Figure 4j*).

Within the mitochondrion, iron is incorporated in proteins of the electron transport chain (*Maclean et al., 2021*). A decrease in mitochondria respiration could be caused by a depletion of proteins in the electron transport chain or the activity of complexes within the electron transport chain. To determine how these are affected by depletion of ZFT, we firstly examined the expression of the key mitochondrial protein succinate dehydrogenase B (SDHB). SDHB has essential roles both in the TCA cycle and in complex II (*Maclean et al., 2021*; *Silva et al., 2023*; *Zwahlen et al., 2024*), contains three Fe–S clusters and has previously been shown to be depleted upon disruption of the mitochondrial Fe–S pathway (*Aw et al., 2021*). We endogenously tagged SDHB (TGME49_215280) in the T7S4-ZFT background (*Figure 4—figure supplement 1f*), confirmed integration by PCR (*Figure 4—figure supplement 1h*), and quantified protein levels following 48 hr of ZFT knockdown. Upon depletion of ZFT, SDHB-3HA expression levels were significantly (p = 0.0205, two-tailed paired *t* test) reduced to around 34%, compared to untreated cells (*Figure 4k, l*). As a control, we also tagged the essential cytosolic Fe–S protein ABCE1 (TGME49_216790) (; *Maclean et al., 2024*; *Figure 4—figure supplement 1g*) and confirmed integration by PCR (*Figure 4—figure supplement 1i*). However, ABCE1-3HA expression levels remained unchanged at 48 hr upon ZFT knockdown (p = 0.1758, two-tailed paired *t* test) (*Figure 4—figure supplement 1j, k*), confirming the lack of a global effect on Fe–S proteins upon ZFT knockdown. We also examined the activity of complex IV, which is predicted to contain both Fe–S proteins and haemoproteins (*Leonard et al., 2023*), by in-gel activity assay (*Lacombe et al.,*

*2019*; *Silva et al., 2023*). We find that ZFT knockdown reduced complex IV activity at 48 hr post ZFT knockdown (*Figure 4m*). Changes in mitochondrial respiration have previously been linked to changes in cristae morphology (*Baker et al., 2019*; *Huet et al., 2018*). Using transmission electron microscopy (TEM) at 72 hr post ATc treatment, we saw no clear changes in cristae morphology, compared to untreated parasites (*Figure 4—figure supplement 1m*).

These data demonstrate ZFT knockdown inhibits apicoplast replication and reduces mitochondrial respiration, potentially due to loss of iron-containing proteins in the mitochondrial electron transport chain.

## Knockdown of ZFT triggers partial parasite differentiation

To determine the effect of ZFT knockdown, we examined parasite replication. Interestingly, replication was not immediately inhibited, and at 48 hr post ATc addition, we observed large vacuoles containing many nuclei (*Figure 5a*), although these large vacuoles did not egress from the host. Around 50% of these vacuoles exhibited loss of normal vacuolar organisation (*Figure 5b*) and we saw frequent examples of asynchronous replication, with only single parasites within a vacuole containing daughter cell scaffolds (*Figure 5a*, **inset**).

To examine the effects of loss of ZFT in more detail, we imaged parasites by TEM. Although there were no obvious morphological changes, at 5 days post knockdown we observed a heavily vesiculated parasitophorous vacuolar membrane (PVM) with the accumulation of small vesicles in the intravacuolar space. In contrast, the PVM of untreated parasites appeared smooth, with no vesicular accumulation (*Figure 5c*). Further, we observed multiple enlarged, electron-lucent vesicles upon ZFT knockdown which were not present in the untreated cells (*Figure 5c*). Similar large vesicles have previously been reported to contain materials that form the cyst wall (*Tomita et al., 2013*), a characteristic of *T. gondii* conversion to bradyzoites. Previously, we and others have demonstrated that loss of iron is sufficient to drive bradyzoite conversion (*Renaud et al., 2024*; *Sloan et al., 2025*; *Ying et al., 2024*). To determine if iron depletion could mimic this phenotype, parental parasites were treated with DFO for 5 days. As with the knockdown of ZFT, iron-depleted parasites also exhibit similar large vesicles close to the PV membrane (*Figure 4—figure supplement 1l*).

To determine if loss of ZFT is sufficient to induce stage conversion, we examined the expression of bradyzoite-specific markers upon ZFT knockdown. BAG1 is a bradyzoite-specific protein (*Kannan et al., 2019*) and DBL is a lectin which binds to the heavily glycosylated cyst wall (*Tomita et al., 2013*). At 72 and 96 hr post ATc addition, we saw vacuoles expressing BAG1 (*Figure 5d*) and with peripheral DBL staining around the vacuole (*Figure 5e*). We quantified this and saw a significantly higher percentage of vacuoles expressing BAG1 and DBL (p = 0.0013 and p = 0.0001, respectively, two-way ANOVA, Tukey corrected for multiple comparisons) upon ZFT knockdown (*Figure 5f, g*). These data show that ZFT depletion leads to the expression of the bradyzoite marker BAG1 and the production of the cyst wall, as detected by DBL.

*T. gondii* differentiation allows cells to survive otherwise fatal stresses, and parasites can reactivate upon removal of the stress (*Cerutti et al., 2020*; *Soete et al., 1993*; *Sullivan and Jeffers, 2012*). To determine if parasites initiating conversion upon ZFT knockdown remained viable, we quantified parasite survival. Parasites were treated with ATc for 72 hr, washed, then incubated without ATc for a further 6 days. After ATc washout, plaques formed, indicating that some parasites remained viable (*Figure 5h*). Upon quantification, around 40% of parasites were able to re-establish growth after 72 hr ZFT knockdown (p < 0.0001, two-tailed unpaired *t* test) (*Figure 5i*), a similar proportion to those expressing the bradyzoite marker BAG1 (*Figure 5f*). These results imply that although many Rh parasites are unable to survive a transient loss of ZFT, around 40% of vacuoles initiate transition to bradyzoites and can survive transient loss of ZFT.

## ZFT knockdown cannot be rescued by excess zinc

As ZIP-domain proteins commonly transport zinc, we next aimed to assess if ZFT also plays a role in zinc uptake in *T. gondii.* Similar to the addition of excess iron, excess zinc was also unable to rescue plaque number (*Figure 6a, b*) or area (*Figure 6c*) in the absence of ZFT. As with iron, changing the promoter alone significantly (p < 0.0001, ordinary one-way ANOVA, Tukey corrected for multiple comparisons) sensitised the parasites to excess zinc, leading to a decrease in plaque size (*Figure 6d*). We also tested a zinc ionophore (2-mercaptopyridine *N*-oxide); however, the zinc ionophore was

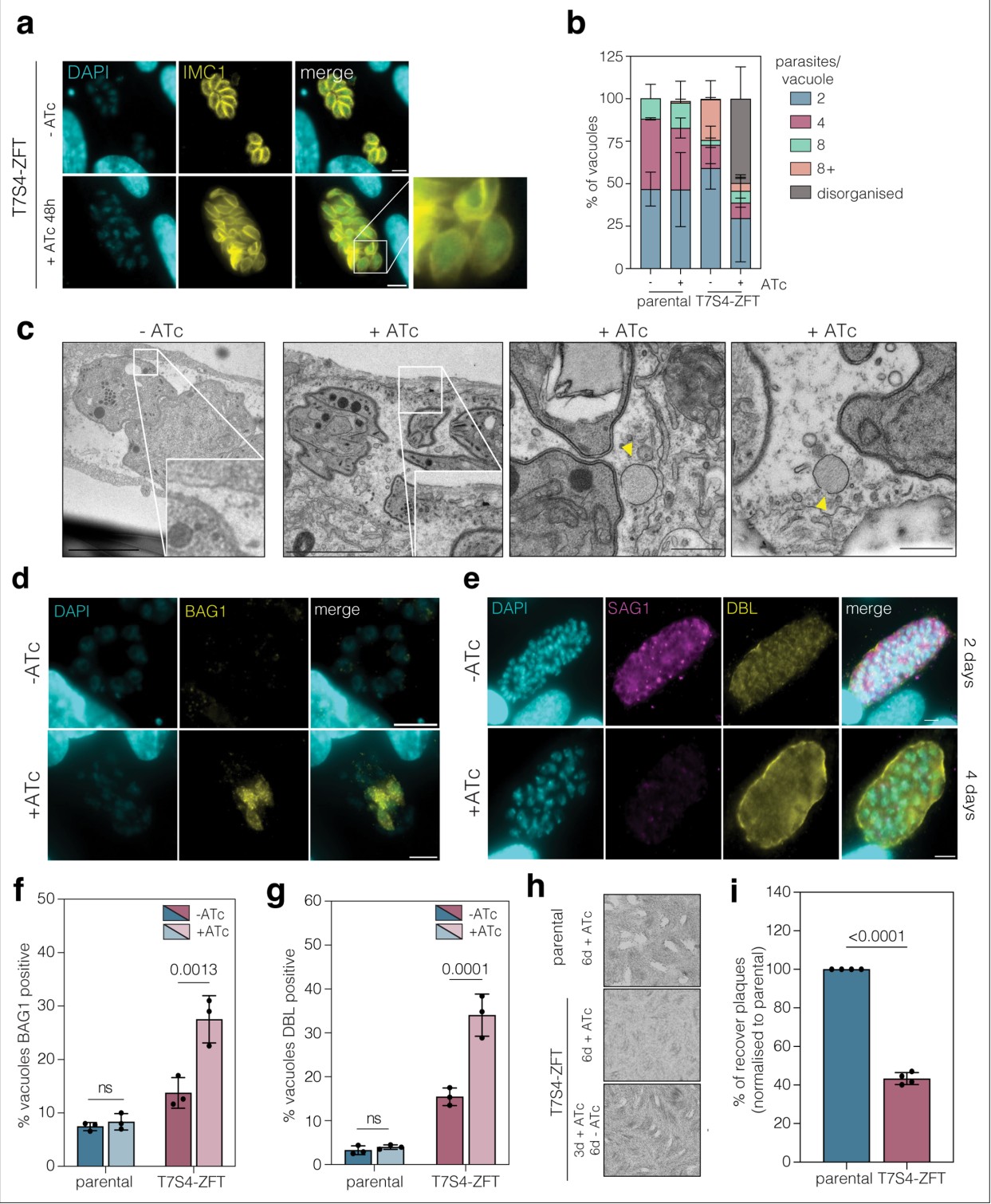

**Figure 5.** Knocking down ZFT triggers partial bradyzoite differentiation. (**a**) Immunofluorescence of T7S4-ZFT parasites uninduced or induced with ATc for 48 hr. Highlighted is a region where parasites are disorganised and asynchronously replicating. Scale bar 5 μm. (**b**) Quantification of the percentage of parasites/vacuole in parental and T7S4-ZFT parasites in the presence and absence of ATc for 48 hr. 100 vacuoles for each condition were counted. Bars at mean of four independent replicates, ± SD. (**c**) TEM images of T7S4-ZFT parasites with and without ATc for 5 days. Detail highlights regions of parasitophorous vacuolar membrane (PVM). Yellow arrows point to enlarged vesicles in the PV space. Scale bar 500 nm. (**d**) Immunofluorescence of T7S4-ZFT-3HA$_{sag1}$ untreated and +ATc 72 hr, stained with BAG1, a bradyzoite specific marker. Scale bar 5 μm. (**e**) Immunofluorescence assay of T7S4-ZFT-3HA$_{sag1}$ untreated and +ATc 4 days, stained with SAG1 and DBL, a cyst wall-binding lectin. Scale bar 5 μm. Quantification of % vacuoles expressing

*Figure 5 continued on next page*

*Figure 5 continued*

BAG1 (**f**) and DBL (**g**). 100 vacuoles for each condition counted, bars at mean of three independent experiments, ± SD. p values from two-way ANOVA, Tukey corrected for multiple comparisons. (**h**) Plaque assays of parental line treated with ATc for 6 days, and T7S4-ZFT parasites, treated either with ATc for 6 days, or for 3 days following washout and 6 days recovery. (**i**) Quantification of plaque number recovery following ATc washout, normalised to parental parasites. Bars at the mean of four independent experiments, ± SD. p values from two-tailed unpaired *t* test.

also unable to rescue the growth phenotype upon knockdown of ZFT (*Figure 6—figure supplement 1a, b*). As we had previously examined the role of iron, we next tested if both excess iron (200 µM FAC) and zinc (25 µM ZnSO₄) were required to rescue the growth phenotype upon ZFT knockdown (*Figure 6a*). However, the addition of both metals was also unable to rescue the T7S4-ZFT growth defect (*Figure 6a and b*). Interestingly, changing the promoter alone leads to a significant (p < 0.0001, two-way ANOVA, Tukey corrected for multiple comparisons) decrease in plaque numbers (to around ~20%) compared to parental parasites in the iron and zinc conditions (*Figure 6—figure supplement 1c*), suggesting that overexpressing ZFT makes the parasites hypersensitive to both iron and zinc. We also tested the sensitivity of the ZFT-Ty overexpression line to zinc. As suggested by the plaque assays, overexpression of ZFT led to a small but significant (p = 0.01, two-tailed unpaired *t* test) increase in zinc sensitivity (*Figure 6e*, *Figure 6—figure supplement 1d*).

We next aimed to understand if the expression of ZFT is dependent on zinc availability. We treated ZFT-3HA$_{zft}$ parasites with 50 µM ZnSO4, however unlike for iron, excess zinc did not alter ZFT-3HA$_{zft}$ expression at 24 hr post treatment (*Figure 6f, g*) and we saw no change in localisation by immunofluorescence (*Figure 6—figure supplement 1e*). We also treated parasites with 5 µM N,N,N',N'-tetrakis-(2-pyridylmethyl) ethylenediamine (TPEN), a cation chelator frequently used to reduce zinc levels (*Schaefer-Ramadan et al., 2019*). TPEN treatment led to a significant (p = 0.04 one-way ANOVA, Tukey corrected for multiple comparisons) reduction in ZFT levels (*Figure 6h, i*). This reduction was somewhat specific to zinc, as it could only be completly rescued by addition of ZnSO4 (*Figure 6h, i*).

## ZFT expression complements lack of zinc transport in *S. cerevisiae*

To investigate whether ZFT has a functional role in zinc uptake in *T. gondii*, we first codon optimised the full length ZFT sequence for expression in *Saccharomyces cerevisiae.* This was then constitutively expressed in either the parental strain (DY1457) or a strain lacking both the low and high affinity zinc transporters, Δ*zrt1/2* (*Ullah et al., 2018*). Under standard conditions, expression of TgZFT complemented the mild growth defect of the Δ*zrt1/2* mutant (*Figure 6j*). Upon chelation of zinc by EGTA, the Δ*zrt1/2* mutant is unable to grow; however, this is rescued by expression of TgZFT (*Figure 6k*). These data demonstrate ZFT expression complements the loss of zinc transport in the Δ*zrt1/2* line, confirming ZFT as a functional zinc transporter.

Based on these results, we also attempted to complement an iron uptake (Δ*fet3/4*) yeast mutant. However, despite multiple attempts, we were unable to express codon-optimised ZFT in the Δ*fet3/4* yeast mutant. The reason for this was not clear, but it is possible that the Δ*fet3/4* yeast strain is less able to survive transformation, or that in this background, expression of ZFT is toxic.

## Knocking down ZFT results in a decrease in parasite-associated metal

To determine if loss of ZFT led to changes in the metal content of parasites, we used the fluorescence iron indicator FerroOrange, which increases fluorescence in the presence of labile iron (*Weber et al., 2020*; *Yu et al., 2022*). To validate its use, we pre-treated parasites with 200 µM FAC intracellularly for 48 hr before live imaging (*Figure 7a*). Parasites preloaded with iron have a significantly (p < 0.0001, two-tailed unpaired *t* test with Welch's correction) higher MFI of FerroOrange compared to untreated (*Figure 7a*)**,** validating the sensitivity of this indicator. Upon 48 hr KD of ZFT, the MFI of FerroOrange was significantly (p = 0.0027, two-tailed unpaired *t* test with Welch's correction) reduced, compared to untreated parasites (*Figure 7b*). We also used the FluoZin-3 indicator which increases in fluorescence in the presence of zinc (*Li et al., 2009*; *Rajapakse et al., 2017*). As above, to validate, we saw that zinc-treated parasites had significantly (p < 0.0001, two-tailed unpaired *t* test with Welch's correction) higher fluorescence than untreated cells (*Figure 7c*). However, upon 24 hr KD of ZFT, the MFI of FluoZin-3AM was significantly (p < 0.0001, two-tailed unpaired *t* test with Welch's correction) reduced compared to untreated parasites (*Figure 7d*), suggesting lower parasite zinc levels in the absence of ZFT.

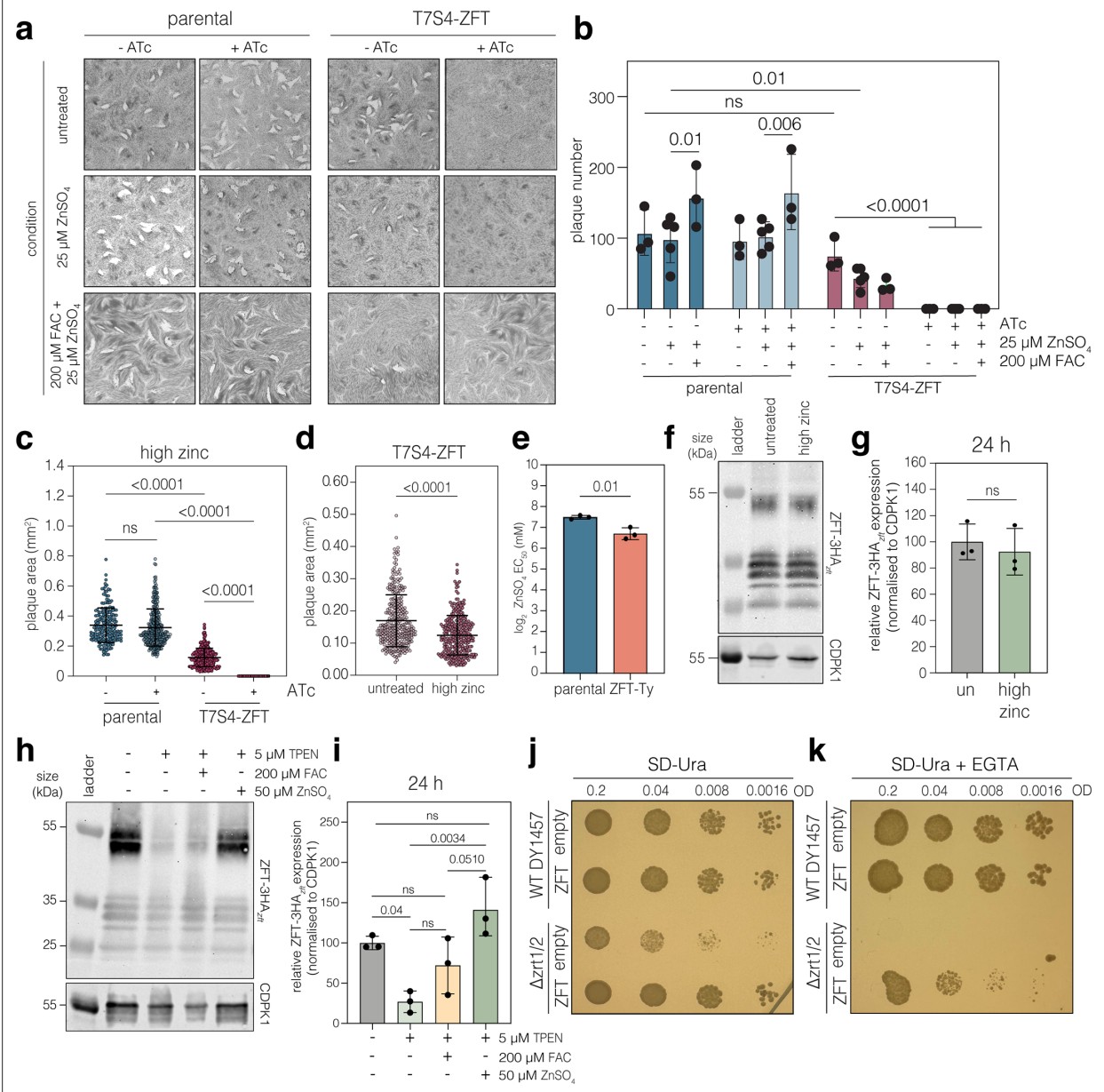

**Figure 6.** ZFT expression depends on the availability of zinc, and ZFT expression complements lack of zinc transport in *S.cerevisiae*. (**a**) Plaque assays showing that the addition of 25 μM ZnSO$_4$ and the combination of 25 μM ZnSO$_4$ and 200 μM FAC cannot rescue ZFT knockdown. (**b**) Number of plaques after treatments. Bars at mean of $n$ = 3, ± SD. p values from two-way ANOVA, Tukey corrected for multiple comparisons. (**c**) Plaque areas of parasites treated with excess zinc (25 μM ZnSO$_4$). Points represent individual plaque areas from two (parental -ATc) or three (all other conditions) independent experiments. Bars at mean ± SD. p values from ordinary one-way ANOVA, Tukey corrected for multiple comparisons. (**d**) Plaque area from T7S4-ZFT parasites in untreated and high zinc (25 μM ZnSO$_4$) conditions. Points represent individual plaque areas from three independent experiments. Bars at mean ± SD. p values are from two-tailed unpaired $t$ test with Welch's correction. (**e**) Graph showing mean EC$_{50}$ for zinc for parental and ZFT-Ty parasites. Bars at mean of $n$ = 3, ± SD. p values from two-tailed unpaired $t$ test. (**f**) Western blot analysis showing ZFT-3HA$_{zft}$ expression levels 24 hr post infection in untreated, high zinc (50 μM ZnSO$_4$). CDPK1 used as a loading control. (**g**) Quantification of ZFT-3HA$_{zft}$ levels at 24 hr from three independent experiments. Bars at mean ± SD. p values from one-way ANOVA, Tukey corrected for multiple comparisons. (**h**) Western blot after treatment with 5 uM TPEN, with additional supplementation by FAC (200 uM) or ZnSO4 (50 uM) as indicated, at 24 hr post infection. (**i**) Quantification of ZFT-3HA$_{zft}$ levels at 24 hr from three independent experiments. Bars at mean ± SD. p values from one-way ANOVA, Tukey corrected for multiple comparisons. (**j**) Spot assay showing TgZFT expression rescues growth in Δzrt1/2 *S. cerevisiae*. Representative of two independent biological experiments. (**k**) Under zinc chelation (5 mM EGTA), TgZFT allows Δzrt1/2 growth. Representative of two independent biological experiments.

The online version of this article includes the following source data and figure supplement(s) for figure 6:

**Source data 1.** PDF file containing original western blots for **Figure 6F, H** indicating the relevant bands and conditions.

*Figure 6 continued on next page*

*Figure 6 continued*

**Source data 2.** Original files for western blot analysis displayed in *Figure 6F, H*.

**Figure supplement 1.** Zinc ionophore is unable to rescue the growth phenotype upon ZFT knockdown, and no change in localisation of ZFT was observed in high iron conditions.

To further examine the effect of knocking down ZFT on parasite-associated metal levels, we performed inductively coupled plasma-mass spectrometry (ICP-MS), an elemental analysis technique used to quantify elemental abundance from complex environments (*Al-Sandaqchi et al., 2018*; *Hare et al., 2012*). At both 24 and 48 hr ATc treatment, we saw no significant changes in the amount of iron in the parasites (*Figure 7e*). However, iron levels were very close to the limit of detection, even in untreated cells, and iron detection by ICP-MS can be challenging due to interference (*Segura et al., 2003*). In contrast, zinc levels were notably higher, and at both 24 and 48 hr ATc treatment, ZFT knockdown parasites had significantly (p = 0.0062 and p = 0.0073, two-tailed paired *t* test) reduced parasite-associated zinc (*Figure 7f*).

Iron can also be quantified at subcellular resolution using X-ray fluorescence microscopy (XFM) (*Aghabi et al., 2023*; *Miller and Ralle, 2024*; *Paunesku et al., 2021*), which allows both spatial and quantitative data. Using XFM, we were able to detect iron in extracellular parasites (*Figure 7g*), which frequently appeared localised to distinct foci in the parasites, which we believe to be the PLVAC, based on our previous observations (*Aghabi et al., 2023*). However, by 48 hr post ATc induction, we were no longer able to detect iron within the parasites. We also assessed zinc localisation. In untreated cells, zinc localised to what we believe is the parasite nucleus (*Figure 7h*), as has been seen in other organisms (*Olea-Flores et al., 2022*). We also saw specific foci of zinc within the cells, which, based on previous results (*Chasen et al., 2019*), we assume to be the PLVAC. Indeed, we found that iron and zinc, along with phosphorus, frequently colocalised in extracellular parasite (*Figure 7i*), as previously has been suggested (*Chasen et al., 2019*). As with iron, depletion of ZFT led to a decrease in parasite-associated zinc, with a loss of zinc both within the nuclei and within foci, which was very evident by 48 hr post knockdown (*Figure 7h*).

Parasite-associated metals were quantified in untreated parasites and at both 24 and 48 hr post ZFT knockdown by XFM. At both timepoints, knockdown parasites had significantly (p < 0.0001, ordinary one-way ANOVA, Tukey corrected for multiple comparisons) less parasite-associated iron, from ~250 nM in untreated, to ~150 nM at 24 hr and ~95 nM at 48 hr (*Figure 7j*). At both 24 and 48 hr KD of ZFT, *T. gondii* parasites have significantly (p < 0.0001, ordinary one-way ANOVA, Tukey corrected for multiple comparisons) less parasite associated zinc, from ~6 µM in untreated, to ~3.5 µM at 24 hr and ~1.7 µM at 48 hr (*Figure 7k*). Interestingly, we found that the concentrations of both sulphur (p = 0.0014, ordinary one-way ANOVA, Tukey corrected for multiple comparisons) (*Figure 7—figure supplement 1a*), and phosphorus (p = 0.008, ordinary one-way ANOVA, Tukey corrected for multiple comparisons) (*Figure 7—figure supplement 1b*), were also significantly reduced at 48 hr following ZFT knockdown, although the magnitude of the decrease was much smaller. Comparison of the changes demonstrated a clear, time-dependent depletion of both iron and zinc, with much smaller drops (around 10–20%) in sulphur and phosphorus (*Figure 7l*). Both sulphur and phosphorus are highly abundant in cellular macromolecules, and this small decrease may speak to defects in protein synthesis or DNA replication upon ZFT knockdown.

## ZFT is an iron and zinc transporter

To confirm that ZFT is sufficient to transport iron, we ectopically expressed ZFT in *Xenopus laevis* oocytes. Surface expression of the transporter was confirmed by streptavidin pull down of biotinylated surface proteins and western blotting (*Figure 8—figure supplement 1a*). To measure if ZFT expression mediated iron uptake, injected oocytes were incubated in the presence of $^{55}Fe^{2+}$ and uptake quantified by scintillation of individual oocytes. We found while control oocytes were unable to take up $^{55}Fe^{2+}$ over time, ZFT expression led to saturable uptake of iron (*Figure 8a*). To determine if ZFT is also able to mediate zinc uptake, injected cells were incubated in the presence of $^{55}Fe^{2+}$, with and without an excess (10x) of unlabelled zinc. Incubation in the presence of zinc strongly (p < 0.0001, one-way ANOVA, Tukey corrected) prevented labelled iron uptake (*Figure 8b*), as previously observed for the human ZIP8 and ZIP14 transporters (*Pinilla-Tenas et al., 2011*; *Wang et al., 2012*).

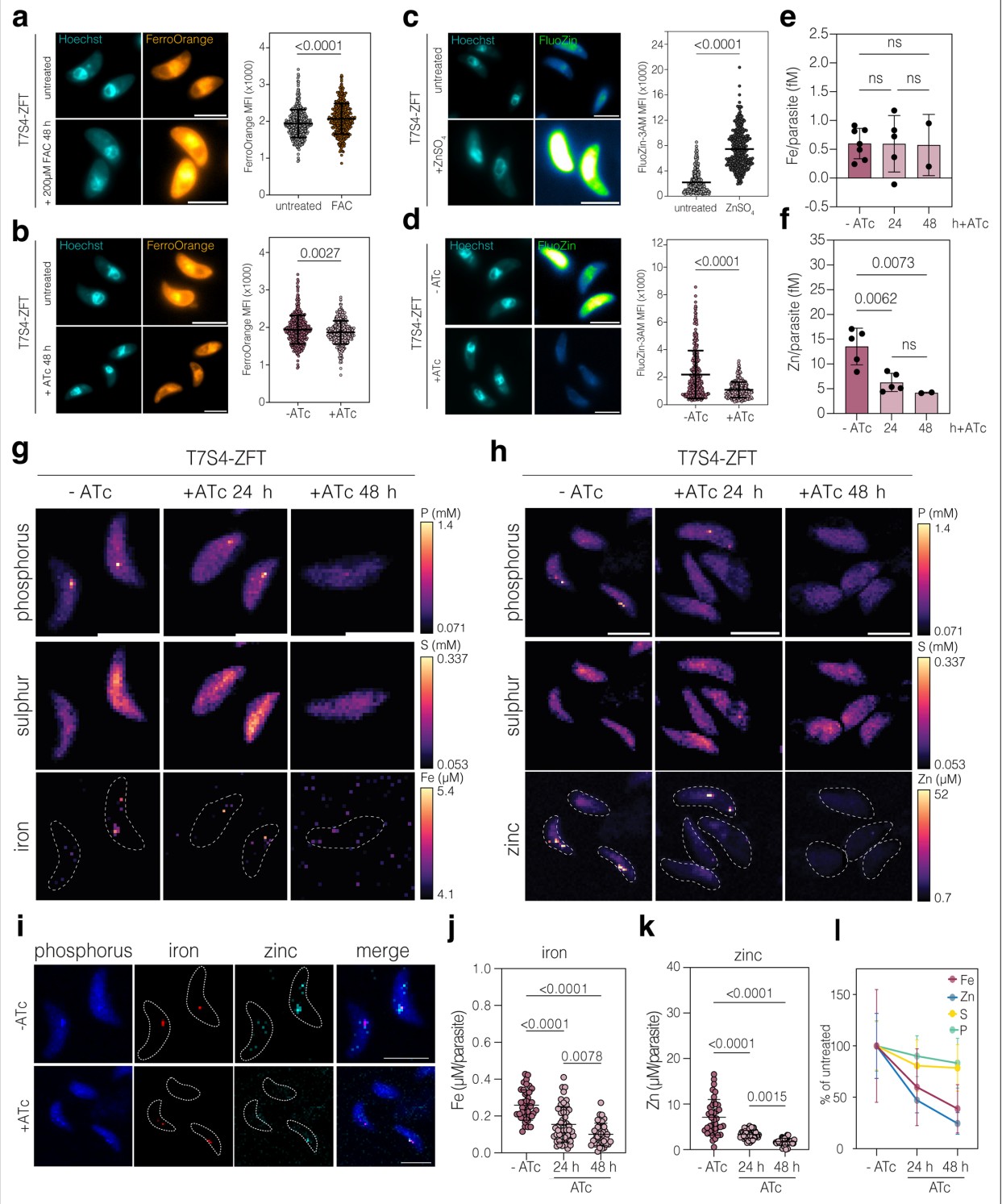

**Figure 7.** Knocking down ZFT results in a decrease in parasite-associated metal. (**a**) Live cell imaging and quantification of FerroOrange stained parental parasites, untreated or treated with 200 μM FAC for 48 hr. Scale bar 5 μm. Points represent the mean fluorescence intensity (MFI) of FerroOrange/ parasite from four independent experiments. Bars at mean ± SD. p value from two-tailed unpaired *t* test with Welch's correction. (**b**) Live cell imaging and quantification of FerroOrange-stained T7S4-ZFT parasites with and without ATc for 48 hr. Points are MFI of FerroOrange/parasite from four independent experiments. Bars at mean ± SD. p value from two-tailed unpaired *t* test with Welch's correction. (**c**) Live cell imaging and quantification of FluoZin-3AM stained parental parasites treated with 50 μM ZnSO₄ for 24 hr. Points represent MFI of FluoZin-3AM/parasite from three independent experiments. Bars at mean ± SD. p value from two-tailed unpaired *t* test with Welch's correction. (**d**) Live cell imaging and quantification of FluoZin-3AM stained T7S4-ZFT parasites with and without ATc for 48 hr. Scale bar 5 μm. Points represent the MFI of FluoZin-3AM per parasite from three independent

*Figure 7 continued*

experiments. Bars at mean ± SD. p value from two-tailed unpaired *t* test with Welch's correction. (**e**) Inductively coupled plasma-mass spectrometry (ICP-MS) quantification of iron/parasite (fM) from T7S4-ZFT parasites with and without ATc at 24 and 48 hr. Bars at mean (*n* = 7 for – ATc, *n* = 5 for +ATc 24 hr and *n* = 2 for +ATc 48 hr) ± SD. p values from one-way ANOVA, Tukey corrected for multiple comparisons. (**f**) ICP-MS quantification of zinc/parasite (fM) from T7S4-ZFT parasites with and without ATc 24 and 48 hr. Bars at mean (*n* = 5 for – ATc and +ATc 24 hr and *n* = 2 for +ATc 48 hr) ± SD. p values from one-way ANOVA, Tukey corrected for multiple comparisons. X-ray fluorescence microscopy (XFM) of extracellular T7S4-ZFT parasites, untreated and treated for 24 and 48 hr, showing phosphorus, sulphur, iron (**g**) and zinc (**h**). (**i**) XFM of untreated parasites showing iron and zinc colocalisation, along with phosphorus. Quantification of Fe (**j**) and Zn (**k**) following ZFT knockdown. Bars at mean ± SD. p values from one-way ANOVA, Tukey corrected for multiple comparisons. (**l**) Quantification of changes in Fe, Zn, S, and P as a percentage of untreated cells, following the addition of ATc. Bars at mean ± SD. Scale bar for all images 5 μm.

The online version of this article includes the following figure supplement(s) for figure 7:

**Figure supplement 1.** Quantification of X-ray fluorescence microscopy (XFM) levels of P (mM) (**a**) and S (mM) (**b**) per parasite following ZFT knockdown at both 24 and 48 hr ATc.

This result, combined with the heterologous result in yeast above, confirms the ability of ZFT to mediate both iron and zinc uptake.

Previous studies using human ZIP transporters have suggested co-transport of iron with bicarbonate, leading to no net change in membrane potential upon iron or zinc transport (*Gaither and Eide, 2000*; *He et al., 2006*; *Pinilla-Tenas et al., 2011*). To test this hypothesis using ZFT, we utilised the electrophysiological two-electrode voltage-clamp (VC) technique, as described previously (*Böhmer et al., 2005*; *Rajendran et al., 2017*). Unlike for human ZIP transporters, addition of $Fe^{2+}$ to oocytes expressing ZFT resulted in an inward current that reached a maximum of –1.2 mA (mean –0.71 mA ± SD 0.21 mA, *n* = 11) (*Figure 8c*). On removal (washout) of iron from the medium, the inward current returned to the baseline values. In contrast, the addition of iron to uninjected oocytes resulted in a much smaller (0.082 mA, mean –0.053 mA ± SD 0.034 mA, *n* = 11) current, consistent with ZFT mediating the electrogenic uptake of iron.

We then defined current–voltage relationships applying VC steps to various potentials between –100 and +50 mV in uninjected and ZFT-expressing oocytes in the presence and absence of $Fe^{2+}$ (*Figure 8d*, representative traces in *Figure 8—figure supplement 1b*). Inward and outward currents were much larger in ZFT-expressing than in uninjected oocytes, but only in the presence of iron. Transport of iron led to reversal of membrane potential at higher voltage (3.66 mV) in ZFT expressing oocytes, compared with uninjected (–44.32 mV untreated, –18.4 mV in $Fe^{2+}$ treated). From the nearly linear relationship between current and voltage, in the range from –50 to +50 mV under all conditions, we calculated conductances (*G* = DI/DV) and found addition of iron in the presence of ZFT led to a conductance of 19 mS SEM 0.067 mS, while conductance in other conditions was significantly lower (uninjected 3 mS, SEM 0. 017 mS, uninjected+$Fe^{2+}$ 6.5 mS, SEM 0.004 mS, ZFT 2.9 mS, SEM 0.018 mS), confirming the transport of iron by ZFT leads to a change in electrogenic membrane potential.

Together, these data validate the role of ZFT as the major iron and zinc transporter of *T. gondii*.

## Discussion

Here we have identified and characterised the first ZIP-domain containing protein in *Toxoplasma*, demonstrating that ZFT is the major iron and zinc importer for the parasite. Sequence analysis of ZFT shows conservation of essential metal coordinating residues and identified a new motif, conserved in coccidia, in the M1 metal binding domain, although with an unknown role in transport.

ZIP proteins mediate metal transport both across the plasma membrane and from vesicles into the cytosol. The localisation of ZFT appears highly dynamic through the parasite's intracellular replication, moving from the plasma membrane to basally clustered intracellular vesicles after two to three rounds of replication. This vacuole stage-dependent localisation is highly unusual in *Toxoplasma* and has not previously been observed. Unfortunately, fluorescent tags led to significant miss-localisation of ZFT to the ER (not shown), so we were unable to observe this trafficking in live cells. Currently, our working model is that ZFT is most active at the plasma membrane, enabling rapid metal incorporation into new vacuoles. As parasite concentrations rise, ZFT is then internalised and later degraded, presumably to prevent toxic metal overload (*Aghabi et al., 2023*). This may explain the persistence of plasma-membrane-localised ZFT in iron starvation and its degradation in excess iron. However, from our

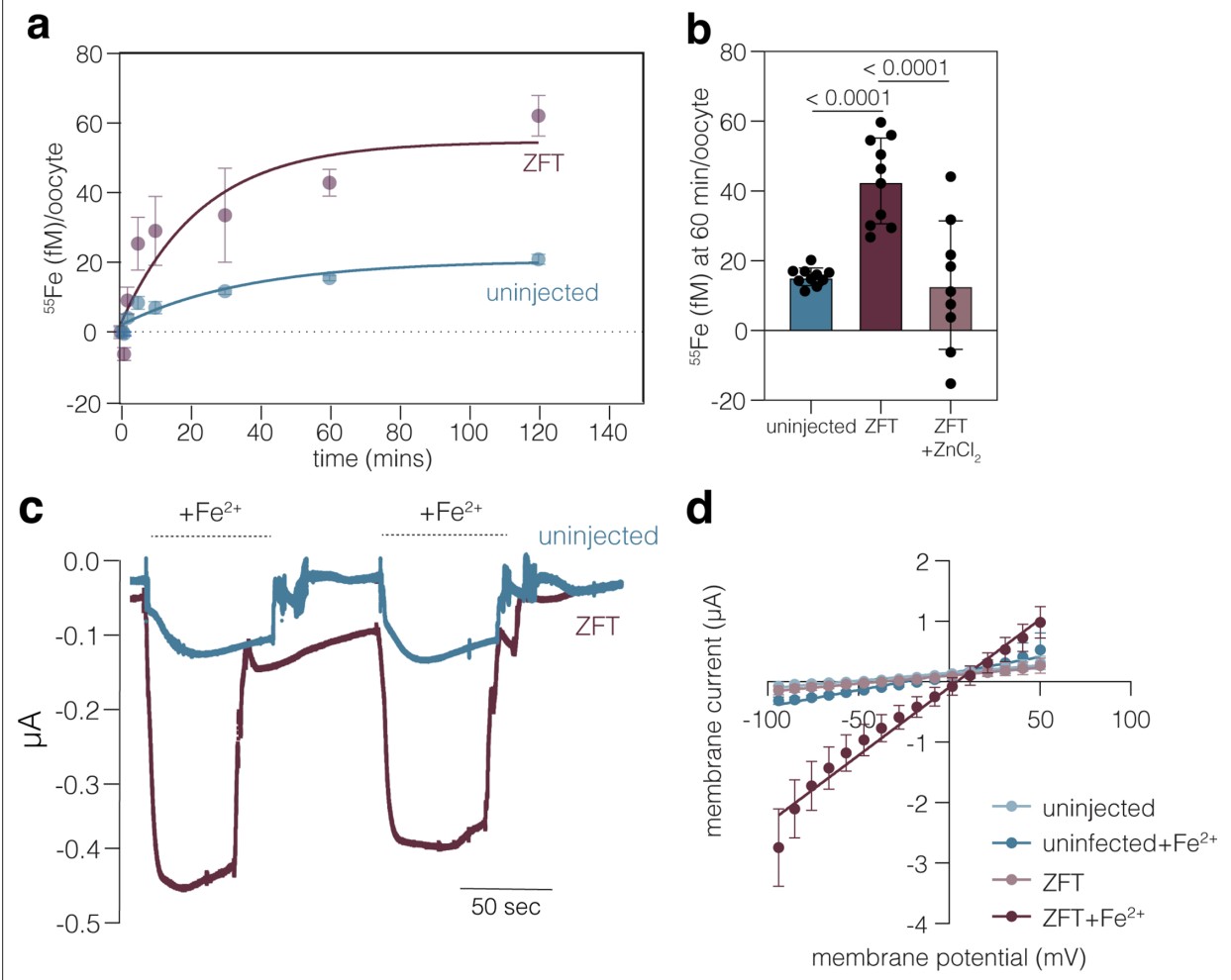

**Figure 8.** Expression of ZFT in *Xenopus laevis* oocytes mediates iron uptake. (**a**) Time course for the uptake of $^{55}Fe^{2+}$ into *X. laevis* oocytes, uninjected or expressing ZFT. Uptake was measured in the presence of 1 mM ascorbic acid to maintain iron in the reduced state. Each data point represents the mean uptake in 10 oocytes from a single experiment, and the data is representative of 3 independent experiments, points and fitted using a first-rate equation. (**b**) $^{55}Fe^{2+}$ uptake at 60 mins, with or without competition with 25 μM $ZnCl_2$. Each bar represents the mean iron uptake of 10 oocytes from a single experiment. (**c**) Representative current record by two-electrode voltage clamp in oocytes expressing or not ZFT after the addition of $Fe^{2+}$ (added as 3 mM $FeCl_3$ with 1 mM ascorbic acid) and a holding potential of –30 mV. (**d**) Current/voltage relationship in control and ZFT expressing oocytes, in the presence and absence of 3 mM $Fe^{2+}$. Values mean of 11 oocytes, ± SD and slopes calculated from simple linear regression.

The online version of this article includes the following figure supplement(s) for figure 8:

**Figure supplement 1.** Western blot of ZFT, and raw electrophysiology traces.

current results, we cannot exclude the hypothesis that the parasite is taking up extracellular material in digestive vesicles, and ZFT transports metals freed from digested proteins into the cytosol. Future studies examining the kinetics of metal uptake, and the localisation dynamics of ZFT in intracellular parasites in more detail may be able to address this remaining question.

In *Toxoplasma*, vacuolar iron (*Aghabi et al., 2023*) and zinc (*Chasen et al., 2019*) transporters have been identified with roles in metal storage and detoxification. However, ZFT is the first *Toxoplasma* metal uptake transporter identified and characterised. This transporter is essential, highlighting the essential role of iron and zinc to the parasite. Interestingly, and unlike in *Plasmodium* (*Loveridge and Sigala, 2024*; *Sahu et al., 2014*), we were unable to complement the loss of this transporter by addition of exogenous metals. One possible explanation could be lack of metal access to the PV; however, our metal dye experiments and the growth phenotype after treatment increase our confidence that exogenous metals reach the vacuole. A more compelling argument is that ZFT is both essential and sufficient for iron and zinc transport, and in its absence, there are no other transporters

able to complement for its loss. As there are three other predicted ZIP transporters in *Toxoplasma*, this raises the questions of their functions within the cell, which will be examined in future studies.

Knockdown of ZFT triggers differentiation into bradyzoites, which complements previous studies demonstrating a similar phenotype upon iron chelation (*Renaud et al., 2024*; *Ying et al., 2024*). The pathway of iron-starvation triggered differentiation is not yet known, although it is compounded by the action of BFD1 (*Sloan et al., 2025*). Interestingly, loss of ZFT leads to a significant inhibition of mitochondrial respiration, with the depletion of the essential, iron-bound protein SDHB (*Maclean et al., 2021*), and reduction of complex IV activity. Inhibition of mitochondrial Fe-S biogenesis, or mitochondrial respiration, has previously been linked to bradyzoite differentiation (*Pamukcu et al., 2021*; *Tomavo and Boothroyd, 1995*); however, we do not yet know the signalling factors linking iron, zinc or mitochondrial function to bradyzoite differentiation. Interestingly, although we see no change in apicoplast-localised lipoylation, we do see an increase in lipoylation of mitochondrial BCDH-E2. Mitochondrial lipoic acid is scavenged from the host cell (*Crawford et al., 2006*), and it is possible that this is linked to the increased reliance on scavenging seen upon iron depletion (*Hanna et al., 2025*). Iron is known to be required in the apicoplast (*Renaud et al., 2022*), zinc also may be required, as the fitness-conferring *Plasmodium* zinc transporter ZIP1 is transiently localised to the apicoplast (*Shrivastava et al., 2024*), although the functional relevance of this localisation has not yet been established. Multiple bioinformatic searches using ToxoDB did not identify a role for zinc in this organelle, although a potential role for zinc cannot be excluded. We see a delayed phenotype on the apicoplast upon ZFT depletion, suggesting that metal import may also be required in this organelle in *Toxoplasma*. However, given the delayed phenotype typically seen upon apicoplast disruption, we cannot determine if this is a direct effect of ZFT, or a downstream consequence of metal depletion. While the use of iron within the parasite has been previously examined, the role of zinc in *Toxoplasma* is largely unknown, although over 150 proteins have predicted Zn-binding domains. XFM suggests a concentration of zinc within the nucleus, where it is expected to play a role in DNA replication and gene expression (*Olea-Flores et al., 2022*). Beyond the nucleus, zinc is clearly concentrated within one or two foci within the cell, in agreement with zinc storage in the PLVAC, as previously hypothesised (*Chasen et al., 2019*). Future studies will be needed to unpick both the specific roles of zinc in *Toxoplasma* and the parasite's response to zinc starvation, as these are likely highly heterogeneous.

ZFT appears to be highly regulated by both high and low iron availability, at the mRNA (*Sloan et al., 2025*), protein and localisation levels. However, ZFT is not regulated by excess zinc, although zinc-mediated ZIP-protein degradation has been observed in other organisms (e.g. *Pang et al., 2023*). Interestingly, there is evidence from other organisms that excess zinc modulates both iron uptake (*Blaby-Haas and Merchant, 2013*; *Xu et al., 2019*) and copper homeostasis (*Xu et al., 2019*). It is possible that excess zinc leads to a decrease in iron acquisition, stabilising ZFT expression; however, this has not yet been assessed. Previously, we have shown that ZFT is stabilised under iron starvation (*Sloan et al., 2025*); however, we did not see the same effect after treatment with the zinc chelator TPEN. There has not yet been any investigation into potential zinc-mediated regulation in *Toxoplasma*, and so we are currently unable to say if this is due to a lack of zinc sensing and response in these parasites, or the treatment or timings we used was insufficient to trigger such a pathway.

The use of XFM to quantify parasite-associated metals has been vital in this work. Beyond localising iron, zinc and phosphorus to an overlapping compartment in extracellular parasites, we were able to show that knockdown of ZFT led to significant decreases in parasite-associated iron and zinc. There are some limitations of XFM, for example we were unable to quantify copper or manganese using this approach, probably due to low concentrations within the parasite. Further, for technical reasons, we worked with chemically fixed parasites which may have led to an underestimate of metal concentrations (*Jin et al., 2017*). However, this technique provides direct evidence of changes in the elemental composition of parasites in the absence of ZFT.

Crucially, here we were able to demonstrate direct transport of $Fe^{2+}$ by ZFT through exogenous expression in *Xenopus* oocytes. This uptake was saturable and was outcompeted by zinc, consistent with a dual specificity transporter. Sequence analysis shows ZFT encodes a leucine at position 403, previously associated with broad substrate specificity (*Gyimesi et al., 2019*) which our results support. We also examined how iron transport affected the plasma membrane conductance and found iron transport is electrogenic under these conditions. This is in contrast to previous studies on human ZIP-transporters (*Gaither and Eide, 2000*; *He et al., 2006*; *Pinilla-Tenas et al., 2011*), however, has

been observed for the plant ZIP transporter OsZIP6 (*Pg et al., 2015*). The mechanism of action of this transporter will require future studies; however, this provides direct evidence of transport activity.

In summary, here we characterise the first iron and zinc uptake transporter in *Toxoplasma*. ZFT is indispensable, highlighting the importance of iron and zinc uptake for the parasite. The identification of this transporter opens new avenues for future investigation into the essential metallobiology of these parasites.

# Materials and methods

**Key resources table**

| Reagent type (species) or resource | Designation | Source or reference | Identifiers | Additional information |
|---|---|---|---|---|
| Strain, strain background (*Toxoplasma gondii*) | Parental | PMID:22144892 | | F3,TATiΔHXΔKu80 |
| Strain, strain background (*Toxoplasma gondii*) | ZFT-3HAzft | PMID:39899594 | TGGT1_261720 | F3,TATiΔHXΔKu80/ZFT-3HAzft |
| Strain, strain background (*Toxoplasma gondii*) | ZFT-3HAsag1 | PMID:39899594 | TGGT1_261720 | F3,TATiΔHXΔKu80/ZFT-3HAsag1 |
| Strain, strain background (*Toxoplasma gondii*) | Parental (tdTomato) | PMID:38162577 | TGGT1_261720 | F3,TATiΔHXΔKu80/tdTomato/ |
| Strain, strain background (*Toxoplasma gondii*) | ZFT-Ty | This paper | TGGT1_261720 | F3,TATiΔHXΔKu80/tdTomato/TUB8-ZFT-Ty-DHFR |
| Strain, strain background (*Toxoplasma gondii*) | T7S4-ZFT | This paper | TGGT1_261720 | F3,TATiΔHXΔKu80/T7S4-ZFT |
| Strain, strain background (*Toxoplasma gondii*) | T7S4-ZFT-3HAsag1 | This paper | TGGT1_261720 | F3,TATiΔHXΔKu80/T7S4-ZFT-3HAsag1 |
| Strain, strain background (*Toxoplasma gondii*) | T7S4-ZFT/SDHB-3HA | This paper | TGGT1_261720; TGGT1_215280 | F3,TATiΔHXΔKu80/T7S4-ZFT/SHDB-3HAsag1 |
| Strain, strain background (*Toxoplasma gondii*) | T7S4-ZFT/ABCE1-3HA | This paper | TGGT1_261720; TGGT1_216790 | F3,TATiΔHXΔKu80/T7S4-ZFT/ABCE1-3HAsag1 |
| Strain, strain background (*Saccharomyces cerevisiae*) | DY1457 | PMID:30258092 | pDR195 | AddGene, #36028 |
| Strain, strain background (*Saccharomyces cerevisiae*) | DY1457 | This paper | pDR195-ZFT | DY1457 expressing ZFT, see methods for details |
| Strain, strain background (*Saccharomyces cerevisiae*) | Δzrt1/2 | PMID:30258092 | pDR195 | AddGene, #36028 |
| Strain, strain background (*Saccharomyces cerevisiae*) | Δzrt1/2 | This paper | pDR195-ZFT | Δzrt1/2 expressing ZFT, see methods for details |
| Cell line (Human foreskin fibroblast) | HFF | ATCC | SCRC-1041 | RRID:CVCL_3285 |
| Sequence-based reagent | All primer and gRNA sequences available in **Tables 1 and 2** | This paper | | See **Tables 1 and 2** |
| Antibody | rat monoclonal anti-HA | Merck 11867423001 | RRID:AB_390918 | 1:500 |
| Antibody | mouse monoclonal anti-Ty | PMID:8813669 | | 1:1000 |
| Antibody | rabbit polyclonal anti-CPN60 | PMID:19808683 | | 1:2000 |
| Antibody | rabbit polyclonal anti-lipoic acid | Abcam ab58724 | RRID:AB_880635 | 1:1000 |

*Continued on next page*

*Continued*

| Reagent type (species) or resource | Designation | Source or reference | Identifiers | Additional information |
|---|---|---|---|---|
| Antibody | guinea pig monoclonal anti-CDPK1 | PMID:31955846 | | 1:10,000 |
| Antibody | rabbit polyclonal anti-rat HRP | Abcam, ab6845 | RRID:AB_955449 | 1:5000 |
| Antibody | goat polyclonal anti-mouse HRP | Promega, W4021 | RRID:AB_430834 | 1:10,000 |
| Antibody | goat polyclonal anti-mouse IRDye 680 | Licor, 926-68070 | RRID:AB_10956588 | 1:10,000 |
| Antibody | goat polyclonal anti-guinea pig IRDye 680 | Thermo Fisher Scientific, SA5-10098 | RRID:AB_2556678 | 1:10,000 |
| Antibody | rabbit polyclonal anti-GAP45 | PMID:18312842 | | 1:1000 |
| Antibody | rabbit polyclonal anti-IMC1 | PMID:12117945 | | 1:1000 |
| Antibody | rabbit polyclonal anti-BAG1 | PMID:31387907 | | 1:1000 |
| Antibody | rabbit polyclonal anti-TOM40 | PMID:27458014 | | 1:1000 |
| Commercial assay or kit | RNeasy Mini Kit | QIAGEN, 74104 | | |
| Commercial assay or kit | High-Capacity cDNA Reverse Transcription Kit | Applied Biosystems, A48571 | | |
| Commercial assay or kit | 4–16% BN-PAGE gel | Thermo Fisher Scientific, BN1002BOX | | |
| Chemical compound, drug | Hoechst 33342 | Thermo Fisher Scientific, H3570 | | |
| Chemical compound, drug | FerroOrange | Dojindo, F374-10 | | |
| Chemical compound, drug | FluoZin-3AM | Thermo Fisher Scientific, F14201 | | |
| Chemical compound, drug | PowerSYBR green PCR master mix | Thermo Fisher Scientific 4367659 | | |
| Chemical compound, drug | 2-Mercaptopyridine *N*-oxide sodium salt (zinc ionophore) | Sigma-Aldrich, H3261 | | |
| Chemical compound, drug | Atovaquone | Thermo Fisher Scientific, 17575545 | | |
| Chemical compound, drug | Carbonyl cyanide 4-(trifluoromethoxy) phenylhydrazone (FCCP) | Merck, C2920 | | |
| Software, algorithm | Prism 10 | GraphPad | | |
| Software, algorithm | FIJI | PMID:22743772 | | |
| Software, algorithm | PyMca 5.9.2 | *Solé et al., 2007* | | |
| Software, algorithm | Clampex 14 | Molecular Devices, CA, USA | | |

## *T. gondii* and host cell maintenance

*T. gondii* tachyzoites were grown in human foreskin fibroblasts (HFFs) cultured in Dulbecco's modified Eagle's medium (DMEM), supplemented with 3% (D3) heat-inactivated foetal bovine serum, 2 mM L-glutamine and 50 U/ml penicillin/streptomycin and maintained at 37°C with 5% $CO_2$.

**Table 1.** List of primer sequences used in this study.

**Primer sequences**

| Name | Sequence (5′–3′) | Notes |
|------|------------------|-------|
| P1 | ctcgttggcatttttcttgATGGCGACGGTCGCAGAC | ZFT amplification from cDNA Fw |
| P2 | tgagcacaacggtgattaatttaatcgagcgggtcctggttcgtgtggacctcAGCGTCTAGAGCCAGATCCAC | ZFT amplification from cDNA and Ty tag Rv |
| P3 | CTCGGTCTTACCCTGGTGGATCTGGCTCTAGACGCaGAGGTCCACACGAACCAGGACCCGCTCGATAGCGGAGCTACTAACTTCAGC | P2A-DHFR amplification Fw |
| P4 | GTGATTAATTTAATCGAGCGGGTCCTGGTTCGTGTGGACCgctaGACAGCCATCTCCATCTGGAT | P2A-DHFR amplification Rv |
| P5 | ctcgttggcatttttcttgatggcgacggtcgcagac | TUB8 ZFT-Ty DHFR integration verification Fw |
| P6 | tgagcacaacggtgattaatttaatcgagcgggtcctggttcgtgtggacctcagcgtctagagccagatccac | TUB8 ZFT-Ty DHFR integration verification Rv |
| P7 | accgaacagggcgacgatttttccccttgagaccttcctCTTCGCCAGGCTGTAAATCC | ZFT T7S4 Fw |
| P8 | agctggcagcagacatcacatcgtctgcgaccgtcgccatTGGTTGAAGACAGACGAAAGCAGTTG | ZFT T7S4 Rv |
| P9 | cctgataggttctggacctccctcgc | ZFT T7S4 5′ integration Fw |
| P10 | cgtagagcagaaacgcactactaaag | ZFT T7S4 5′ integration Rv |
| P11 | gcaaagccagagggatatatgctcc | ZFT T7S4 3′ integration Fw |
| P12 | cgaggtcgacgtatcgataagcttacg | ZFT T7S4 3′ integration Rv |
| P13 | AAATAGGATGGTACATCCCCC | HA integration test Rv |
| P14 | gaaacacacttcgtgcagc | CAT integration test Fw |
| P15 | CTTTTCTTTTGTGGGTTGTGC | SDHB-3HA Fw |
| P16 | CGACATTTGCATGTACATACC | SDHB-3HA UTR Rv |
| P17 | ATCTCGTCAGCGGCATGAACC | ABCE1-3HA Fw |
| P18 | CATACACGGACACAGATACGC | ABCE1-3HA UTR Rv |
| P19 | aaaaaaaatataccccagcctcgagATGGCCACCGTCGCTGATGAC | Amplification of ZFT for yeast expression Fw |
| P20 | aagtccaaagctggatccgcctcgagTTAGTCAAGAGGGTCTTGGTTAG | Amplification of ZFT for yeast expression Rv |
| P21 | GGGACGACATGGAGAAAATC | Actin RT-qPCR Fw |
| P22 | AGAAAGAACGGCCTGGATAG | Actin RT-qPCR Rv |
| P23 | CGGACAACCGAATAGTGGCT | ZFT RT-qPCR Fw |
| P24 | CGCCAACGATCATTCCTTCG | ZFT RT-qPCR Rv |
| P25 | cttttggcagatcaattccccaccatgGCGACGGTCGCAGACGATGTGATG | ZFT open reading frame amplification from RHΔKu80 cDNA template Fw |
| P26 | ccgggacatcgtacgggtacgaAGCGTCTAGAGCCAGATCCACCAGG | ZFT open reading frame amplification from RHΔKu80 cDNA template Rv |

## *T. gondii* strain construction

ZFT-3HA$_{zft}$ and ZFT-3HA$_{sag1}$ parasite strains were generated as previously described (*Sloan et al., 2025*). To generate RHtdTomato pTUB8 ZFT-Ty-DHFR parasites, ZFT was amplified from cDNA and a Ty tag was inserted at the 3′ end of the gene by PCR using primers P1 and P2 (*Table 1*). The tubulin promoter

**Table 2.** A list of the gRNA sequences used in this study.

**gRNA sequences**

| Name | Sequence (5'–3') |
| --- | --- |
| ZFT 3' | TGAACTGTTGCACAACCTAC |
| ZFT 5' | CCTTGAGACCTTTCCTCCCG |
| SDHB 3' | CGAGCCGAGGTTACGCAGCG |
| ABCE1 3' | GCTCGACGACGCTTGAAGCG |

TUB8 plasmid was digested with EcoRI and PacI and then assembled with the ZFT-Ty PCR product to generate the new plasmid. The DHFR sequence was amplified using primers P3 and P4. The TUB-ZFT-Ty plasmid was then digested using BbcC1 enzyme, and the DHFR sequence was inserted into the plasmid. TUB8-ZFT-Ty-DHFR plasmid was linearised using enzyme ScaI and then transfected into RHtdTomato parasites using a Gene Pulse BioRad electroporator. Parasites were selected using pyrimethamine and cloned by limiting dilution. Clones were confirmed using primers P5 and P6 (**Table 1**). For the generation of the F3ΔHX T7S4-ZFT strain, the T7S4 promoter was amplified from the G13m5 plasmid (**Sheiner et al., 2011**) using primers P7 and P8. F3ΔHX parasites were transfected using a Gene Pulse BioRad electroporator with 30 μg of the purified repair template T7S4-ZFT PCR product and 40 μg of a plasmid containing Cas9 guide targeting the 3' end of the genomic sequence of ZFT (**Table 2**). Parasites were selected with pyrimethamine and cloned by limiting dilution. Clones were screened for 3' integration using primers P9 and P10, and 5' integration using primers P11 and P12. For the generation of the F3ΔHX T7S4-ZFT-3HA$_{sag1}$ line, ZFT was tagged as previously described (**Sloan et al., 2025**) in F3ΔHX T7S4-ZFT parasites, selected for with chloramphenicol, and cloned by limiting dilution. The T7S4-ZFT SDHB-3HA and T7S4-ZFT ABCE1-3HA parasite strains were generated by tagging SDHB and ABCE1 as previously described (**Maclean et al., 2024**; **Maclean et al., 2021**) in the F3ΔHX T7S4-ZFT parasites, selected for with chloramphenicol, and cloned by limiting dilution. T7S4-ZFT SDHB-3HA clones were screened for 5' integration using primers P13 and P15, and 3' integration using primers P14 and P16. T7S-ZFT ABCE1-3HA clones were screened for 5' integration using primers P13 and P17, and 3' integration using primers P14 and P18. Parasite lines were selected using either chloramphenicol (Sigma-Aldrich, C3175), made up in MiliQ water and used at 40 μg/ml or pyrimethamine (Sigma-Aldrich, 46706), made up in ethanol and used at 2 μM. All sgRNA sequences (**Table 2**) and primers (**Table 1**) used in the study are listed below.

## Chemicals

Ferric ammonium citrate (FAC) (Sigma-Aldrich, F5879) and Deferoxamine mesylate salt (DFO) (Sigma-Aldrich, D9533), were made up in 1X phosphate-buffered saline (PBS) and used as indicated. Zinc sulphate (ZnSO$_4$) (Sigma-Aldrich, 307491) and 2-mercaptopyridine *N*-oxide sodium salt (Sigma-Aldrich, H3261) were made up in distilled water and used at indicated concentrations. TPEN (Merck, 616394) was made up in DMSO and used at 5 μM. Anhydrotetracycline hydrochloride (ATc) (abcam, ab145350), made up in ethanol and used at 0.5 μM, unless indicated.

## Immunofluorescence assays

IFAs were performed on infected HFF monolayers. Intracellular *T. gondii* parasites were fixed with 4% paraformaldehyde (PFA) for 20 min at room temperature and washed with 1X PBS. Cells were permeabilised and blocked in blocking buffer (PBS/0.2% Triton X-100/2% bovine serum albumin) for 30 min at room temperature, or at 4°C overnight. Coverslips were then incubated in a wet chamber with primary antibodies rat anti-HA 1:500 (Merck 11867423001), rabbit anti-GAP45 (**Plattner et al., 2008**) 1:2000, a kind gift from Dominique Soldati-Favre, rabbit anti-IMC1 (**Wichroski et al., 2002**) 1:2000, rat anti-SAG1 1:500 (Abcam), rabbit anti-BAG1 (**Kannan et al., 2019**; **Smith et al., 2021**) 1:500, a gift from Lilach Sheiner, rabbit anti-TOM40 (**van Dooren et al., 2016**) 1:1000, a kind gift from Dr. Giel Van Dooren (ANU) and rabbit anti-CPN60 (**Agrawal et al., 2009**) 1:1000, a kind gift from Lilach Sheiner, in blocking buffer for 1 hr at room temperature. Coverslips were then washed three times with PBST (0.2% Triton X-100 in PBS). After washing, slides were incubated in a wet chamber with the secondary antibodies Goat anti-rat AlexaFluor-594 1:1000 (Invitrogen), goat anti-mouse AlexaFluor-488 1:1000

(Invitrogen), goat anti-rabbit AlexaFluor-488 1:1000 (Invitrogen), goat anti-guinea pig-488 1:1000 (Invitrogen) or Dolichos Biflorus-488 (DBL) 1:1000, in blocking buffer for 1 hr at room temperature in the dark. Coverslips were then washed with PBST (0.2% Triton X-100 in PBS), and mounted using Fluoromount with 4',6-diamidino-2-phenylindole, to stain DNA (Southern Biotech, 0100-20).

### FerroOrange live cell imaging

To analyse the levels of iron in the parasites, FerroOrange (Dojindo, F374-10) was used. Parasites were untreated, treated with ATc or 200 µM FAC for 48 hr. Parasites were released, pelleted and resuspended in DMEM supplemented with 1 µM FerroOrange and 10 µM Hoechst 33342 (ThermoFisher Scientific, H3570), put onto poly-L-lysine treated live cell dishes (Cellvis, D29-20-1.5-N) and incubated at 37°C for 10 min before imaging. Images were obtained using an inverted DiM8 microscope (Leica Microsystems) using LAS X Core software and processed using FIJI software. Intensity was quantified using FIJI (*Schindelin et al., 2012*).

### FluoZin-3AM live cell imaging

To analyse the levels of zinc in the parasites, a $Zn^{2+}$ selective indicator FluoZin-3AM (Thermo Fisher Scientific, F14201) was used. Parasites were untreated, treated with ATc or 50 µM $ZnSO_4$ for 24 hr. Parasites were released, pelleted and resuspended in DMEM supplemented with 2 µM FluoZin-3 and 10 µM Hoechst 33342 (Thermo Fisher Scientific, H3570), put onto poly-L-lysine treated live cell dishes (Cellvis, D29-20-1.5-N) and incubated at 37°C for 10 min before imaging. Images were obtained using an inverted DiM8 microscope (Leica Microsystems) using LAS X Core software and processed using FIJI software. Intensity was quantified using FIJI (*Schindelin et al., 2012*).

### RT-qPCR

To validate the T7S4-ZFT parasites, RT-qPCR was performed to confirm successful knockdown of ZFT mRNA levels. HFFs were infected with T7S4-ZFT with and without ATc for 24, 48, 72, and 96 hr before being mechanically lysed and filtered through a 3-mm filter to remove any host cell debris. Parasites were then harvested by centrifugation at 2000 × *g* for 10 min, supernatant was removed, pellet was washed once in sterile PBS before spinning down in a microcentrifuge at 7000 × rpm for 10 min. Supernatant was removed and pellets were frozen at –20°C overnight. RNA was then extracted from the pellets using the RNeasy Mini Kit (QIAGEN, 74104). 1 µg RNA was then treated with DNAse I 1 U/µl (Thermo Fisher Scientific, 18068015) at room temperature for 15 min before inactivating the DNAse with 25 mM EDTA followed by DNase denaturation at 65°C for 10 min. Using 1 mg of DNase I treated RNA, cDNA synthesis was then performed using the High-Capacity cDNA Reverse Transcription Kit (Applied Biosystems, A48571) according to the manufacturer's instructions. RT-qPCR was carried out on the Applied Biosystems 7500 Real Time PCR machine using PowerSYBR green PCR master mix (Thermo Fisher Scientific 4367659) and 2 ng of 1:10 diluted cDNAs per reaction. Primers P21 and P22 were used for actin as a housekeeping control gene and primers P23 and P24 were used for ZFT (*Table 1*). The results are from three independent biological experiments; each performed in technical triplicate. Relative fold changes for - ATc vs. +ATc samples were calculated using the Pfaffl method (*Pfaffl, 2001*). GraphPad Prism 10 was used to perform statistical analysis.

### Transmission electron microscopy

Infected cells were fixed with 2.5% glutaraldehyde and 4% PFA in 0.1 M cacodylate buffer. Following serial washes in 0.1 M cacodylate buffer, the material was post-fixed in 1% $OsO_4$ and 1.25% $K4[Fe(CN_6)]$ for 1 hr in the dark, and contrasted en bloc with 0.5% aqueous uranyl acetate for 1 hr at room temperature in the dark. The samples were then dehydrated in ascending acetone series and embedded in epoxy resin. Ultra-thin sections (60 nm) were collected using a Leica Ultramicrotome UC6 and observed in a Jeol 1200EX transmission electron microscope (Jeol, Japan).

### Plaque assays

200, 500 or 1000 parental or ZFT-Ty/T7S4-ZFT parasites were applied onto confluent monolayers of HFF cells in a 6-well plate in D3 media, supplemented when indicated with 200 µM FAC, 25 µM $ZnSO_4$, 2 µM zinc ionophore (2-mercaptopyridine *N*-oxide sodium salt (Sigma-Aldrich, H3261)), or both 200 µM FAC and 25 µM $ZnSO_4$, in the presence or absence of 0.5 µM ATc, unless stated otherwise.

Parasites were allowed to replicate for 7 days undisturbed, washed 3x with PBS, fixed with ice cold 100% methanol and stained using crystal violet stain (12.5 g crystal violet in 125 ml ethanol, diluted in 500 ml 1% ammonium oxalate) for 2–3 hr at room temperature before washing with distilled $H_2O$, drying and imaging. For ATc washout experiments, ATc was removed from cells after 3 days. Cells were then washed 2x with PBS and fresh D3 media was added to the cells. Parasites were allowed to replicate for a further 6 days undisturbed, then fixed and stained as described above.

## Fluorescent growth assays

Growth assays of fluorescent parasites were performed as previously (*Aghabi et al., 2023*). Briefly, host cells were infected with 1000 either RHtdTomato (*Elati et al., 2023*) or RHtdTom TUB8 ZFT-Ty tachyzoites per well and allowed to invade for 2 hr at 37°C. The media was then removed and supplemented with either FAC, DFO, $ZnSO_4$, or Ars (at indicated concentrations) and incubated at 37°C with 5% $CO_2$ for 4 days. Total parasite fluorescence was recorded at 4 days using a PHERAstar FS microplate reader (BMG LabTech). All experiments were performed in triplicate with at least three independent biological replicates. Uninfected wells served as blanks and fluorescence values were normalised to infected, untreated wells. Dose-response curves and $EC_{50}$ were calculated using GraphPad Prism 10.

## Western blotting

Parasites were mechanically released, filtered and collected by centrifugation at $1500 \times g$ for 10 min, and lysed with RIPA lysis buffer (150 mM sodium chloride, 1% Triton X-100, 0.5% sodium deoxycholate, 0.1% sodium dodecyl sulphate (SDS) and 50 mM Tris, pH 8.0) for 30 min on ice. Samples were then resuspended with 5X Laemmli buffer (10% SDS, 50% glycerol, 300 mM Tris-HCl pH 6.8 and 0.05% bromophenol blue), boiled at 95°C for 5 (unless stated otherwise) and separated on a 10% SDS–PAGE gel, initially at 120 V for 20 min then 175 V for 60 min. PageRuler Prestained Protein Ladder (Thermo Fisher Scientific, 26619) was used as a molecular weight marker. Proteins were then wet transferred to nitrocellulose membrane (0.2 µM, Cytiva, 10600004) in Towbin buffer (0.025 M Tris, 0.192 M glycine, 10% methanol) for 60 min at 250 mA and blocked for at least 1 hr at room temperature in blocking buffer (5% milk in 0.1% Tween/PBS). Blots were then stained with primary antibodies overnight at 4°C rat anti-HA, Merck 11867423001 1:500, mouse anti-Ty (*Bastin et al., 1996*) 1:1000, rabbit anti-CPN60 (*Agrawal et al., 2009*) 1:2000, a kind gift from Lilach Sheiner, rabbit anti-lipoic acid, Abcam ab58724, 1:1000 and guinea pig anti-CDPK1 (*Waldman et al., 2020*) 1:10,000, both kind gifts from Sebastian Lourido, followed by secondary fluorescent antibodies at room temperature for 1 hr (rabbit anti-rat HRP 1:5000 (abcam, ab6845), goat anti-mouse HRP 1:10,000 (Promega, W4021), goat anti-mouse IRDye 680 1:10,000 (Li-Cor) and goat anti-guinea pig IRDye 680 1:10,0000 (Li-Cor)). ZFT-3HA$_{zft}$ was detected using Pierce ECL Western Blotting Substrate (ThermoFisher Scientific) and imaged using the Invitrogen iBright 1500 Imaging System. CDPK1 was detected using the LI-COR Odyssey LCX system.

For Blue Native (BN) PAGE, $10^7$ parasites were mechanically released, filtered, and harvested by centrifugation at $1500 \times g$ for 10 min. The parasite pellet was then resuspended in a BN lysis buffer (1% DDM, 750 mM aminocaproic acid, 0.5 mM EDTA and 50 mM Bis-Tris-HCl pH 7.0), and incubated on ice for 30 min before being centrifuged at $16,000 \times g$ for 30 min at 4°C. The supernatant containing the solubilised membrane proteins was then added to a sample buffer containing coomassie G250 (NativePAGE), with the final concentration being 0.0625% Coomassie G250 and 0.25% DDM. $10^7$ parasites were loaded onto a 3–12% Native PAGE Bis-Tris gel (Thermo Fisher Scientific, NP0301PK2). NativeMark was used as the molecular marker. Proteins were then transferred onto a PVDF membrane (0.45 µM, Cytiva, 10600023) via wet transfer in Towbin buffer for 60 min at 100 V. Immunoblotting and chemiluminescence detection was performed as above.

## Complex IV in gel oxidation assay

Intracellular *T. gondii* tachyzoites were harvested by centrifugation at $1500 \times g$ for 10 min at 4°C. The parasites were washed once with 1x PBS and centrifuged at $12,000 \times g$ for 1 min at 4°C. Parasite pellets were then resuspended in BN-PAGE lysis buffer (25% (wt/vol) 4x Native PAGE buffer (Invitrogen, BN20032), 2% (wt/vol) protease inhibitors, 1% (vol/vol) digitonin (Sigma-Aldrich, D141) in distilled water) to a final concentration of $2.5 \times 10^5$ parasites/ml. Samples were kept on ice for 30 min then spun at $20,000 \times g$ for 30 min at 4°C. The supernatant was transferred to a fresh labelled tube.

15 ml of sample per well was aliquoted, and ¼ of G250 additive (5% stock) as the detergent % in the sample was added. Samples were run on a 4–16% BN-PAGE gel (Novex-Life technologies) in the cold room at 4°C with the dark cathode buffer in the centre of the rank at 150 V for 60 min. The dark cathode buffer was then replaced with the light cathode buffer and run for a further 90 min at 250 V until the dye had reached the bottom of the gel. To perform the in-gel enzyme stain, the oxidation buffer (50 mM OF $KH_2PO_4$ pH 7.2 in distilled water) was prepared. 1 mg/ml cytochrome c (Sigma-Aldrich, C7752) and 0.1% (wt/vol) 3'3-diaminobenzidine tetrahydrochloride (Sigma-Aldrich, D5637) were added to the oxidation buffer. The gel was then gently removed from the plastic cassette and incubated in the staining solution overnight before imaging.

## Seahorse assay/extracellular flux analysis

The *Toxoplasma* Seahorse assay was performed as previously described (*Hayward et al., 2022*). Briefly, parasites were incubated in the absence or presence of ATc for 48 hr, extracellular parasites were removed by washing 1X PBS, and intracellular parasites were mechanically released, filtered and harvested by spinning at 2000 × *g* for 10 min. Parasites were then resuspended in Seahorse XF media (Agilent) supplemented with 5 mM glucose and 1 mM glutamine at 37°C. $1.5 \times 10^6$ parasites in a final volume of 175 µl were added to each well of a poly-L-lysine-treated seahorse XFp cell culture miniplate (Agilent). Parasites were then centrifuged at 300 × *g* for 5 min to ensure sedimentation before the basal OCR and the ECAR were quantified using a Seahorse XF HS Mini Analyser (Agilent Technologies). The OCR following the addition of 1 µM atovaquone (Scientific, 17575545) was considered as the non-mOCR and subtracted as background. The basal and maximal OCRs of the parasites were calculated by subtracting the background from the raw OCR values before and after the addition of 1 µM carbonyl cyanide 4-(trifluoromethoxy) phenylhydrazone (FCCP) (Merck, C2920), respectively. Basal ECAR values were taken as raw values. This experiment was performed in three independent biological replicates, each with three technical triplicates.

## ICP-MS

Total parasite-associated iron and zinc were quantified using ICP-MS. Confluent HFF cells in T150 flasks were infected with T7S4-ZFT parasites for 24 hr before the addition of ATc for a further 24 hr or infected with T7S4-ZFT parasites +ATc for 48 hr to ensure efficient knockdown. Untreated (-ATc) parasites were used as a control. Parasites were harvested by scraping, syringing and filtering through either 0.3 µM filters or PD-10 columns (Cytiva, 17085101), and quantified using a haemocytometer. Parasites were then washed three times with PBS supplemented with 6% Chelex 100 sodium form (Merck, C7901), then once with 1 mM EDTA before being digested in 65% nitric acid (Merck, 225711) at 85°C for 5 hr. Digested samples were then diluted 1:10 in 2% nitric acid with 0.025% Triton X-100. At least three biological replicates were collected, with three technical replicates per biological sample. Samples were quantified at the Scottish Universities Environmental Research Centre (SUREC) and processed by Dr Valerie Olive. Briefly, samples were run on an Agilent 7500ce ICP-MS, fitted with a double pass spray chamber and a self-aspire nebuliser tuned for 0.1 ml/min. A 3 ppb Indium ($^{115}$In) was introduced as an internal standard to correct for the sensitivity drop caused by the sample matrix. Cu and Zn were taken on the masses 63 and 66, respectively, and the results are an average of 10 repeats. The calibration equation was calculated from 3 points at 5, 10, and 20 ppb using Cu and Zn mono-element solutions Cu and Zn specpure . Iron was taken at the mass 56 using a collision cell with a flow of hydrogen at 4 ml/min to suppress the interference of the $^{40}Ar^{16}O$. The calibration equation was calculated using a 200 ppb and 300 ppb solution of mono-element Fe specpure, results are an average of 20 repeats.

## X-ray fluorescence microscopy

T7S4-ZFT parasites were treated in the presence or absence of ATc for 24 or 48 hr. Parasites from a fully lysed dish were filtered, pelleted, and resuspended in 5 ml of extracellular buffer (116 mM NaCl, 5.4 mM KCl, 0.8 mM $MgSO_4$, 5.5 mM glucose, 50 mM HEPES, pH 7.2). Approximately 500 µl of resuspended parasites were then added onto each poly-L-lysine 2 × 2 mm silicon nitride (SiN) window (Norcada, X5200D) and spun down gently at 200 × *g* for 2 min then left at room temperature for 20 min.

Following adhesion/infection, cells were then washed twice with 1X chelexed PBS and fixed in 4% PFA made up in 1X chelexed PBS for 20 min at room temperature. Cells were then washed twice in 1X chelexed PBS, twice in 100 mM ammonium acetate and finally twice in MiliQ $H_2O$ before drying overnight. Elemental mapping was performed at the European Synchrotron Radiation Facility in Grenoble, France at beamline ID21 under proposal number LS-3419. Data are available from https://doi.esrf.fr/10.15151/ESRF-DC-2119497225. Analysis was performed using PyMca 5.9.2 (*Solé et al., 2007*) and FIJI (*Schindelin et al., 2012*). The 48 hr time point is representative of two independent biological experiments, whereas the 24 hr time point is representative of one independent biological experiment. We would like to thank Hiram Castillo Michel for assistance and support.

## Yeast strain construction

Full length ZFT was cloned into the yeast expression vector pDR195 (AddGene, #36028) using primers P19 and P20 (*Table 1*). Wildtype (DY1457) and Δzrt1/2 (*Ullah et al., 2018*) strains, a kind gift from Jim Dunwell (University of Reading), were transformed as previously described (*Gietz and Schiestl, 2007*). Positive transformations were selected on synthetic dropout agar lacking uracil (SD-Ura). Colonies were grown overnight in SD-Ura media and spotted onto SD-Ura agar, or SD-Ura agar supplemented with 5 mM EGTA and incubated for 4 days.

## *X. laevis* oocyte preparation and *Tg*ZFT expression

The open reading frame of *Tg*ZFT was amplified from RHΔ*Ku80* strain cDNA template using the primers P25 and P26. The resultant product was cloned by Gibson assembly into the vector pGHJ-HA previously digested with *Xma*I and *Avr*II enzymes (*Rajendran et al., 2017*). The plasmid was linearised by incubation with *Not*I overnight, and complementary RNA (cRNA) encoding HA-tagged *Tg*ZFT was prepared for injection into oocytes as previously described (*Bröer, 2010*; *Fairweather et al., 2012*; *Rajendran et al., 2017*; *Wagner et al., 2000*). *X. laevis* oocytes were surgically removed and prepared for cRNA injection as described (*Bröer, 2010*). For all oocyte experiments, 20 ng of *Tg*ZFT cRNA was micro-injected into stage 5 or 6 oocytes using a Micro4 micro-syringe pump controller and A203XVY nanoliter injector (World Precision Instruments). *X. laevis* frogs were maintained and used in accordance with protocols approved by the Australian National University Animal Ethics Committee (A2023/24).

## Oocyte surface biotinylation and whole membrane preparation

Oocyte surface biotinylation was performed as described previously (*Bröer, 2010*; *Fairweather et al., 2015*). Briefly, for surface biotinylation, five oocytes were selected 5 days post cRNA injection, washed three times with ice-cold PBS (pH 8.0), incubated for 10 min at room temperature in 0.5 mg/ml of EZ-Link Sulfo-NHS-LC-Biotin (Thermo Fisher Scientific), and then washed four times with ice-cold PBS. Oocytes were subsequently solubilised in oocyte lysis buffer (20 mM Tris-HCl pH 7.6, 150 mM NaCl, 1% vol/vol Triton X-100) for 2 hr on ice. Samples were centrifuged for 15 min at 16,000 × g, and the supernatant was mixed with 50 µl of streptavidin-coated agarose beads (Thermo Fisher Scientific) and incubated overnight at 4°C to purify surface exposed proteins. Beads were washed four times with oocyte lysis buffer before elution in 20 µl of SDS–PAGE sample buffer with 4% b-mercaptoethanol. Protein samples from surface biotinylation were separated by SDS–PAGE and ZFT-HA detected by western blotting using anti-HA, as described above.

## Oocyte $^{55}$Fe uptake

Uptake experiments were conducted as described previously (*Fairweather et al., 2015*) with the following specification for $^{55}$Fe as the uptake substrate. Briefly, 10 non-injected or 4 days post cRNA injection oocytes were washed four times with ND96 buffer (96 mM NaCl, 2 mM KCl, 1 mM MgCl$_2$, 1.8 mM CaCl$_2$, 10 mM MES, pH 5.5) at RT and then incubated in 200 µl of labelling solution (2.7 µM $^{55}$FeCl, 2 mM ascorbic acid in ND96 pH 5.5). For the Zn competition assay, labelling solutions were supplemented with 25 µM ZnCl$_2$. Uptake was quenched by washing oocytes four times with 1 ml of ice-cold ND96. After uptake, single oocytes were distributed into scintillation vials and lysed by addition of 30 µl of 10% SDS. 2 ml of Microscint-40 scintillation fluid (Perkin-Elmer) was added to the samples and radioactivity was counted on a Hidex 300 SLL scintillation counter. An experiment was performed in three independent replicates.

## Electrophysiological recordings in *X. laevis* oocytes

Electrical recordings of iron-induced responses were performed as described previously (*Fairweather et al., 2012*; *Rajendran et al., 2017*). All recordings were performed using a TURBO TEC-03X solid state amplifier (npi electronic, Tamm, Germany) in VC mode with a potential headstage of x 10 mV sensitivity, output impedance of 50 $\Omega$ and output voltage of ±15 V. The x 1 current headstage was used in VC mode with a full bandwidth of 10 Hz, current range of 150 µA/1 M$\Omega$ and voltage range of ±150 V. Electrodes were encased by Borosilicate glass capillaries (WPI, Sarasota, FL, USA) and filled with 3 M KCl solution over the silver chloride filaments. The electrodes were then tested and used at between 0.5 and 1.3 M$\Omega$ resistance. Data acquisition was performed using a Molecular Devices Digidata 1550B series driven by Clampex 14. Software (Molecular Devices, CA, USA). Raw data is publicly available here: https://doi.org/10.17605/OSF.IO/NGC5W. In all electrophysiology experiments where substrate was applied multiple times, the baseline of current tracings was zero in the absence of substrate perfusion. This was done to quantify the subsequent current generated by application of substrate minus the background current of the oocyte. Oocytes were initially impaled with the potential end current electrode in clamp-free mode to register a healthy membrane potential of > –25 mV. Following the zeroing of offset currents and elimination of any current bias, VC mode was engaged and set to a holding potential of –30 mV for all recordings. Continuous trace recordings of ZFT-injected and uninjected oocytes were conducted using a gap-free protocol with application of substrate manually to the recording bath.

Single oocytes were placed in a 1 ml chamber with ND96 (pH 5.5) buffer, impaled in two-VC configuration and allowed to recover to a steady-state $E_m$ before recordings began. Only oocytes with basal $E_m$ between –30 and –20 mV were used. All recordings were conducted in ND96 (pH 5.5) buffer. Before voltage clamping, the amplifier output current was set to zero to normalise currents recorded in VC mode. Oocytes were clamped at –30 mV unless indicated otherwise. Fe$^{+2}$ (added as FeCl$_3$ into 1 mM L-ascorbic acid) was added to a final concentration of 3 mM, in ND96 (pH 5.5) to assess current changes.

## Acknowledgements

DA is supported by the Wellcome IIB PhD programprogramme (218518/Z/19/Z) and EG is supported by the BBSRC NorthWestBio PhD programprogramme (BB/X010902/1). This work was supported by the BBSRC (BB/W014947/1) to LP and CRH. CRH is funded by a Sir Henry Dale Fellowship from the Wellcome Trust and the Royal Society (213455/Z/18/Z). SJF and GGvD are supported by an Australian Research Council Discovery Project (DP230100853). We acknowledge the European Synchrotron Radiation Facility (ESRF) for provision of synchrotron radiation facilities under proposal numbers LS-3200 and LS-3419, and we would like to thank Hiram Castillo-Michel for assistance and support in using beamline ID21. The authors gratefully acknowledge the MVLS Cellular Analysis facility for their support and assistance and https://www.ToxoDB.org/, without which this work would not have been possible. The funders had no role in study design, data collection and analysis, decision to publish, or preparation of the manuscript.

## Additional information

### Funding

| Funder | Grant reference number | Author |
| --- | --- | --- |
| Wellcome | 10.35802/218518 | Dana Aghabi |
| Wellcome | 10.35802/213455 | Clare R Harding |
| Australian Research Council | DP230100853 | Stephen J Fairweather Giel G van Dooren |
| Biotechnology and Biological Sciences Research Council | BB/X010902/1 | Erin J Gibson |

| Funder | Grant reference number | Author |
|---|---|---|
| Biotechnology and Biological Sciences Research Council | BB/W014947/1 | Lucas Pagura<br>Clare R Harding |

The funders had no role in study design, data collection, and interpretation, or the decision to submit the work for publication. For the purpose of Open Access, the authors have applied a CC BY public copyright license to any Author Accepted Manuscript version arising from this submission.

## Author contributions

Dana Aghabi, Conceptualization, Data curation, Formal analysis, Investigation, Methodology, Writing – original draft, Writing – review and editing; Cecilia Gallego Rubio, Investigation, Methodology; Miguel Cortijo Martinez, Augustin Pouzache, Erin J Gibson, Investigation; Lucas Pagura, Investigation, Methodology, Writing – original draft, Writing – review and editing; Stephen J Fairweather, Giel G van Dooren, Resources, Methodology; Clare R Harding, Conceptualization, Data curation, Formal analysis, Supervision, Funding acquisition, Investigation, Writing – original draft, Writing – review and editing

## Author ORCIDs

Dana Aghabi (ID) https://orcid.org/0000-0002-7827-5296
Erin J Gibson (ID) https://orcid.org/0009-0009-7648-1318
Lucas Pagura (ID) https://orcid.org/0000-0003-1061-6311
Stephen J Fairweather (ID) https://orcid.org/0000-0003-1530-7876
Giel G van Dooren (ID) https://orcid.org/0000-0003-2455-9821
Clare R Harding (ID) https://orcid.org/0000-0002-8428-0374

Reviewer #1 (Public review): https://doi.org/10.7554/eLife.108666.3.sa1
Reviewer #2 (Public review): https://doi.org/10.7554/eLife.108666.3.sa2
Reviewer #3 (Public review): https://doi.org/10.7554/eLife.108666.3.sa3
Author response https://doi.org/10.7554/eLife.108666.3.sa4

---

# Additional files

## Supplementary files
MDAR checklist

## Data availability

Elemental mapping was performed at the European Synchrotron Radiation Facility in Grenoble, France at beamline ID21 under proposal number LS-3419. Data is are available from https://doi.org/10.15151/ESRF-DC-2119497225. Electrophysiology data acquisition was performed using a Molecular Devices Digidata 1550B series driven by Clampex 14. Software (Molecular Devices, CA, USA). Raw data is publicly available here: https://doi.org/10.17605/OSF.IO/NGC5W.

The following datasets were generated:

| Author(s) | Year | Dataset title | Dataset URL | Database and Identifier |
|---|---|---|---|---|
| Perry C, Aghabi D, Pagura L, Gibson E | 2025 | Iron and zinc in extracellular *Toxoplasma gondii* | https://doi.org/10.15151/ESRF-DC-2119497225 | European Synchrotron Radiation Facility, 10.15151/ESRF-DC-2119497225 |
| Harding C | 2025 | Electrophysiology of Xenopus oocytes expressing *Toxoplasma gondii* ZFT | https://doi.org/10.17605/OSF.IO/NGC5W | Open Science Framework, 10.17605/OSF.IO/NGC5W |

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
