## [Editor Report · eLife Assessment]

This **important** study identifies a metal transporter in the plasma membrane of the obligate intracellular pathogen, *Toxoplasma gondii*. Using an array of different approaches, the authors **convincingly** demonstrate that this transporter mediates iron and zinc uptake and regulates diverse cellular processes, including parasite metabolism and differentiation. This work will be of broad interest to cell biologists and biochemists studying metal ion transport mechanisms.

---

## [Referee Report · Reviewer #1 (Public review)]

In this manuscript, Aghabi et al. present a comprehensive characterization of ZFT, a metal transporter located at the plasma membrane of the eukaryotic parasite *Toxoplasma gondii*. The authors provide convincing evidence that ZFT plays a crucial role in parasite fitness, as demonstrated by the generation of a conditional knock-down mutant cell line, which exhibits a marked impact on mitochondrial respiration, a process dependent on several iron-containing proteins. Consistent with previous reports, the authors also show that disruption of mitochondrial metabolism leads to conversion into the persistent bradyzoite stage.

The study then employed advanced techniques, such as inductively coupled plasma-mass spectrometry (ICP-MS) and X-ray fluorescence microscopy (XFM), to demonstrate that ZFT depletion results in reduced parasite-associated metals, particularly iron and zinc. Additionally, the authors show that ZFT expression is modulated by the availability of these metals, although defects in the transporter could not be compensated by exogenous addition of iron or zinc. Finally, the authors used heterologous expression of ZFT in Xenopus oocytes and yeast mutants, highlighting the dual substrate specificity of the transporter. The ability of ZFT to transport both iron and zinc is thus supported by two experimental approaches in heterologous systems. First by demonstrating ZFT ability to transport zinc, as the expression of Toxoplasma ZFT can compensate for a lack of zinc transport in yeast. Then, by showing the ability of ZFT to transport iron, as assessed in the Xenopus oocytes model. Furthermore, phenotypic analyses suggest defects in iron availability upon ZFT depletion, particularly with regard to Fe-S mitochondrial proteins and mitochondrial function.

Overall, the manuscript provides a solid, well-rounded argument for ZFT's role in metal transport, using a combination of complementary approaches. The converging evidence, including changes in metal concentrations upon ZFT depletion, data on metal transport obtained in heterologous systems, and phenotypic changes linked to iron deficiency, presents a convincing case. Given that metal acquisition remains largely uncharacterized in Toxoplasma, this manuscript provides an important first step in identifying a metal transporter in these parasites, and the data presented are generally convincing and insightful.

Comments on revisions:

The revised manuscript has successfully addressed all of the key points raised in the initial review. Notably, the metal transport experiments in Xenopus oocytes now provide compelling evidence supporting the role of ZFT function. I congratulate the authors on their efforts and have no further concerns to raise.

---

## [Referee Report · Reviewer #2 (Public review)]

Summary:

The intracellular pathogen *Toxoplasma gondii* scavenges metal ions such as iron and zinc to support its replication; however, mechanistic studies of iron and zinc uptake are limited. This study investigates the function of a putative iron and zinc transporter, ZFT. In this paper, the authors provide evidence that ZFT mediates iron and zinc uptake by examining the regulation of ZFT expression by iron and zinc levels, the impact of altered ZFT expression on iron sensitivity, and the effects of ZFT depletion on intracellular iron and zinc levels in the parasite. The effects of ZFT depletion on parasite growth are also investigated, showing the importance of ZFT function for the parasite.

Strengths:

A key strength of the study is the use of multiple complementary approaches to demonstrate that ZFT is involved in iron and zinc uptake. The heterologous expression of ZFT in a Xenopus oocyst system where ZFT was shown to transport iron and zinc is an important addition to the study. The authors also build on their finding that loss of ZFT impairs parasite growth by showing that ZFT depletion induces stage conversion and leads to defects in both the apicoplast and mitochondrion.

Weaknesses:

The inclusion of the data showing iron and zinc transport when ZFT is expressed in a Xenopus oocyst system alleviated one of the main weaknesses of the original paper - the lack of direct biochemical evidence that ZFT acted as an iron transporter.

---

## [Referee Report · Reviewer #3 (Public review)]

Summary:

Aghabi et al set out to characterize a *T. gondii* transmembrane protein with a ZIP domain, termed ZFT. The authors investigate the consequences of ZFT downregulation and overexpression for parasite fitness. Downregulation of ZFT causes defects in the parasite's endosymbiotic organelles, the apicoplast and the mitochondrion. Specifically, lack of ZFT causes a decrease in mitochondrial respiration, consistent with its role as an iron transporter. This impact on the mitochondria appears to trigger partial differentiation to bradyzoites. The authors furthermore demonstrate that expression of TgZFT can rescue a yeast mutant lacking its zinc transporter and perform an array of direct metal ion measurements including X-ray fluorescence microscopy and inductively coupled mass spectrometry (ICP-MS). These reveal reduced metal ions in parasites depleted in ZFT. In the manuscript's revision, the authors performed additional transport assays in Xenopus oocysts, providing further evidence for the transporter trafficking iron. Overall, the data by Aghabi et al. convincingly support that ZFT is a major metal ion transporter in *T. gondii*, importing iron and zinc for diverse essential processes.

Strengths:

This study's strength lies in the thorough characterization of the transporter. The authors combine a number of techniques to measure the impact of ZFT depletion, ranging form the direct measurement of metal ions to determining the consequences for the parasite's metabolism (mitochondrial respiration) as well as performing a yeast mutant complementation and transport assays in Xenopus oocysts expressing the *T. gondii* protein. This work is very thorough and clearly presented, leaving little doubt about this protein's function.

Weaknesses:

None. The authors have addressed all my previous queries/ concerns.

---

## [Author Response]

The following is the authors’ response to the original reviews.

**Public Reviews:**

**Reviewer #1 (Public review):**
In this manuscript, Aghabi et al. present a comprehensive characterization of ZFT, a metal transporter located at the plasma membrane of the eukaryotic parasite *Toxoplasma gondii*. The authors provide convincing evidence that ZFT plays a crucial role in parasite fitness, as demonstrated by the generation of a conditional knockdown mutant cell line, which exhibits a marked impact on mitochondrial respiration, a process dependent on several iron-containing proteins. Consistent with previous reports, the authors also show that disruption of mitochondrial metabolism leads to conversion into the persistent bradyzoite stage. The study then employed advanced techniques, such as inductively coupled plasma-mass spectrometry (ICP-MS) and X-ray fluorescence microscopy (XFM), to demonstrate that ZFT depletion results in reduced parasite-associated metals, particularly iron and zinc. Additionally, the authors show that ZFT expression is modulated by the availability of these metals, although defects in the transporter could not be compensated for by exogenous addition of iron or zinc.While the manuscript does not directly investigate the transport function of ZFT through biochemical assays, the authors indirectly support the notion that ZFT can transport zinc by demonstrating its ability to compensate for a lack of zinc transport in a yeast heterologous system. Furthermore, phenotypic analyses suggest defects in iron availability, particularly with regard to Fe-S mitochondrial proteins and mitochondrial function. Overall, the manuscript provides a solid, well-rounded argument for ZFT's role in metal transport, using a combination of complementary approaches. Although direct biochemical evidence for the transporter's substrate specificity and transport activity is lacking, the converging evidence, including changes in metal concentrations upon ZFT depletion, yeast complementation data, and phenotypic changes linked to iron deficiency, presents a convincing case. Some aspects of the results may appear somewhat unbalanced, particularly since iron transport could not be confirmed through heterologous complementation, while zinc-related phenotypes in the parasites have not been thoroughly explored (which is challenging given the limited number of zinc-dependent proteins characterized in Toxoplasma). Nevertheless, given that metal acquisition remains largely uncharacterized in Toxoplasma, this manuscript provides an important first step in identifying a metal transporter in these parasites, and the data presented are generally convincing and insightful.

We thank the reviewer for their assessment and would like to highlight that we now add direct biochemical characterisation in the new Figure 8, supporting our hypothesis and confirming iron transport by this protein.

**Reviewer #2 (Public review):**
Summary:The intracellular pathogen *Toxoplasma gondii* scavenges metal ions such as iron and zinc to support its replication; however, mechanistic studies of iron and zinc uptake are limited. This study investigates the function of a putative iron and zinc transporter, ZFT. In this paper, the authors provide evidence that ZFT mediates iron and zinc uptake by examining the regulation of ZFT expression by iron and zinc levels, the impact of altered ZFT expression on iron sensitivity, and the effects of ZFT depletion on intracellular iron and zinc levels in the parasite. The effects of ZFT depletion on parasite growth are also investigated, showing the importance of ZFT function for the parasite.Strengths:A key strength of the study is the use of multiple complementary approaches to demonstrate that ZFT is involved in iron and zinc uptake. Additionally, the authors build on their finding that loss of ZFT impairs parasite growth by showing that ZFT depletion induces stage conversion and leads to defects in both the apicoplast and mitochondrion.Weaknesses:(1) Excess zinc was shown not to alter ZFT expression, but a cation chelator (TPEN) did lead to decreased expression. While TPEN is often used to reduce zinc levels, does it have any effect on iron levels? Could the reduction in ZFT after TPEN treatment be due to a reduction in the level of iron or another cation?

WE thank the reviewers for this comment, we agree that TPEN is a fairly unspecific cation chelator so to determine if its effects are due to removal of zinc or other cations we treated with TPEN and either zinc or iron. Co-incubation of TPEN and zinc prevented ZFT depletion, while TPEN+FAC had no effect compared to TPEN alone (new Figure 6h and i), strongly suggesting the effects on ZFT abundance are linked to zinc and not just iron.

(2) ZFT expression was found to be dynamic depending on the size of the vacuole, based on mean fluorescence intensity measurements. Looking at protein levels by Western blot at different times during infection would strengthen this finding.

We show here that ZFT expression is highly dynamic, depending both the iron status of the host cell and the number of parasites/vacuole. However, validating this finding by western would be complex due to the highly unsynchronised nature of parasite replication and the large number (5x10^6^ - 1x10^7^cells) of parasites required to visualise ZFT. Further, we show that ZFT is apparently internalised prior to degradation. For this reason, we have not attempted to validate this finding by western blotting at this time.

(3) ZFT localization remained at the parasite periphery under low iron conditions. However, in the images shown in Figure S1c, larger vacuoles (containing 4-8 parasites) are shown for the untreated conditions, and single parasite-containing vacuoles are shown for the low iron condition. As ZFT localization is predominantly at the basal end of the parasite in larger PV and at the parasite periphery for smaller vacuoles, it would be better to compare vacuoles of similar size between the untreated and low-iron conditions.

The reviewer brings up a good point, the concentration of iron chelator that we used here does not enable parasite replication, making an assessment of changes in localisation challenging. To address this, have new data using a much lower concentration of chelator (20 mM), which is still expected to impact the parasites (Hanna et al, 2025), but allows for replication. In this low iron environment, ZFT localisation remained significantly more peripheral (Fig. S1d,e), supporting our hypothesis that ZFT localisation is iron dependent, independent of vacuolar stage.

**Reviewer #3 (Public review):**
Summary:Aghabi et al set out to characterize a *T. gondii* transmembrane protein with a ZIP domain, termed ZFT. The authors investigate the consequences of ZFT downregulation and overexpression for parasite fitness. Downregulation of ZFT causes defects in the parasite's endosymbiotic organelles, the apicoplast and the mitochondrion. Specifically, lack of ZFT causes a decrease in mitochondrial respiration, consistent with its role as an iron transporter. This impact on the mitochondria appears to trigger partial differentiation to bradyzoites. The authors furthermore demonstrate that expression of TgZFT can rescue a yeast mutant lacking its zinc transporter and perform an array of direct metal ion measurements, including X-ray fluorescence microscopy and inductively coupled mass spectrometry (ICP-MS). These reveal reduced metal ions in parasites depleted in ZFT. Overall, the data by Aghabi et al. reveal that ZFT is a major metal ion transporter in *T. gondii*, importing iron and zinc for diverse essential processes.Strengths:This study's strength lies in the thorough characterization of the transporter. The authors combine a number of techniques to measure the impact of ZFT depletion, ranging from the direct measurement of metal ions to determining the consequences for the parasite's metabolism (mitochondrial respiration), as well as performing a yeast mutant complementation. This work is very thorough and clearly presented, leaving little doubt about this protein's function.Weaknesses:This study offers no major novel insights into the biology of *T. gondii*. The transporter was already annotated as a zinc transporter (ToxoDB), was deemed essential (PMID: 27594426), and localized to the plasma membrane (PMID: 33053376). This study mostly confirms and validates these previous datasets. The authors identify three other proteins with a ZIT domain. Particularly, the role of TGME49_225530 is intriguing, as it is likely fitness-conferring (score: -2.8, PMID: 27594426) and has no subcellular localization assigned. Characterizing this protein as well, revealing its localization, and identifying if and how these transporters coordinate metal ion transport would have been worthwhile.

We agree that the work presented here validates the previous datasets, and if that was all we had done, we agree that the biological insights would be limited. However, we have gone significantly beyond the predictions, demonstrating dynamic localisation changes, iron-mediated regulation, the lack of substrate-based complementation and validating transport activity of both zinc and iron. Although *in silico* predictions and screens can be informative, it remains important to validate biological functions experimentally. While we agree that characterisation of TGME49_225530 (as well as the other two annotated ZIP proteins) would be interesting, and will certainly form part of our future plans, it is significantly beyond the scope of the presented manuscript.

Another weakness is the data related to the impact of ZFT downregulation on the apicoplast in Figure 4. The authors show that downregulation of ZFT causes an increase in elongated apicoplasts (Figure 4d). The subsequent panels seem to show that the parasites present a dramatic growth defect at that time point. This growth arrest can directly explain the elongated apicoplast, but does not allow any conclusion about an impact on the organelle. In any case, an assessment of 'delayed death' as presented in Figure 4c seems futile, since the many other processes affected by zinc and iron depletion likely cause a rapid death, masking any potential delayed death.

To address this point, we agree that given the importance of iron and zinc to the parasite that we cannot differentiate the death of the parasite due to apicoplast defects from death from other causes and we have modified the discussion to reflect this, as below.

“However, given the delayed phenotype typically seen upon apicoplast disruption, we cannot determine if this is a direct effect of ZFT, or a downstream consequence of metal depletion”

**Recommendations for the authors:**

**Reviewer #1 (Recommendations for the authors):**
Specific Comments:(1) The background on the typical sequence features that would identify Toxoplasma ZIP homologues should be expanded and clarified. While these proteins are likely quite divergent and may lack many conserved features, the manuscript currently does not provide enough detail to assess how similar (or different) TgZIPs are from well-characterized family members. Additionally, the justification for focusing on TGGT1_261720 (ZFT) over TGGT1_225530, as stated in the first paragraph of the results section, seems unclear. There is no predictive data supporting a potential plasma membrane localization for TGGT1_225530 (yet this cannot be excluded), and TGGT1_225530 appears to have more canonical metal-binding motifs. I believe that the fact that only TGGT1_261720 is iron-regulated should be sufficient justification for its selection, and this point could be emphasized more clearly. Furthermore, the discussion mentions a leucine residue that may be associated with broad substrate specificity, but this is not addressed in the initial comparative sequence analysis. These residues and the HK motif are not actually addressed in the Gyimesi et al. reference currently mentioned; thus this could be clarified and updated with references (such as PMID: 31914589) that provide more recent insights into key residues involved in metal selectivity in ZIP transporters.

We thank you for this comment, to address these points:

We agree that the iron-mediated regulation is sufficient for our focus on ZFT and have clarified the text to reflect this, as described above.

We have also updated the references as suggested, our apologies for this oversight.

We have further expanded the discussion, especially with reference to our new results using heterologous expression in oocytes (please see above).

(2) Figure 1D, Figure 2A, C, H, Figure 3D, Figure 6F, H, corresponding text and paragraph 2 of the Discussion: It seems that most of the "non-specific bands" annotated in Figure 1D, which are lower molecular weight products, are not present in the parental cell line, suggesting they may not be non-specific after all. These bands also vary depending on the cell line (e.g., promoter used, see Figures 2H and 3D) or experimental conditions (e.g., iron excess or depletion). Given the dynamic localization of ZFT during intracellular development, it may be worth exploring whether these lower molecular weight bands represent degraded forms of TgZFT, possibly corresponding to the basally-clustered signal observed by immunofluorescence, with only the full-length protein associating with the plasma membrane. This possibility should be investigated or at least discussed further.

While the lower bands are not present in the parental, we do see them in other HA-tagged lines, especially when the expression of the tagged protein is low, seen below (Author response image 1). We don’t currently have an explanation for these, but we can confirm that they do not change in abundance in parallel with the full length protein, supporting our hypothesis that these bands are an artefact of the anti-HA antibody in our system. Although ZFT is clearly degraded (e.g. Fig. 1g), we currently do not believe these bands are ZFT c-terminal degradation products.

**Author response image 1. sa4fig1:** Western blot of ZFT-3HA_*zft*_ and another HA-tagged unrelated cytosolic protein, demonstrating that the lower bands are most likely nonspecific.

(3) It is unfortunate that ZFT could not complement a yeast iron transporter mutant cell line, as this would have provided a strong argument for ZFT's role in iron transport. The manuscript does not provide much detail about the Δfet2/3 yeast mutant line. Fet3 is the ferroxidase subunit, while Ftr1 is the permease subunit of the high-affinity iron transport complex in yeast. Fet2, however, appears to be *Saccharomyces cerevisiae*'s VPS41 homolog. Therefore, is Δfet2/3 the most appropriate mutant to use, or would another mutant line (e.g., ΔFtr1) be a better choice? Additionally, while Figure 7 suggests a decrease in metal uptake upon ZFT depletion, it would be useful to test whether overexpression of ZFT leads to enhanced metal incorporation, perhaps using a FerroOrange assay.

We thank the reviewer for their comments, which we have answered below:

The Δfet2/3 yeast mutant was a typo and has been corrected, or apologies, we did use the Δfet3/4 mutant line, based on previous successful experiments involving plant metal transporters (e.g (DiDonato et al., 2004)).

Unfortunately, we were unable to perform the FerroOrange assay in the overexpression line as this line is endogenously fluorescent in the same channel as FerroOrange.

However, as detailed above we have now added significant new data, confirming our hypothesis that ZFT is an iron/zinc transporter through heterologous expression in *Xenopus* oocytes in the new figure 8. This provides direct evidence of transport of iron, and evidence that zinc can inhibit this transport, consistent with our hypothesis.

(4) The annotation of the blot in Figure 2H suggests that overexpressed ZFT-TY can only be detected in the absence of heat denaturation. However, this is not addressed in the text. Does heat denaturation also affect the detection of ZFT-3HA or the lower molecular weight products? This should be clarified in the manuscript.

Interestingly, ZFT is detectable after boiling at 95° C for 5 minutes when expressed at endogenous (or near endogenous) levels in the ZFT-3HA_*sag1*_ and ZFT-3HA_*zft*_ tagged parasite lines. However, overexpression of ZFT leads to a loss of detection via western blot when boiled, although the protein is detectable without heat denaturation.

A possible explanation for this is that overexpression of protein may cause ZFT to miss-fold, making the protein more prone to aggregation following boiling, rendering the protein insoluble and unable to enter the gel. Moreover, heat aggregation can sometimes mask the epitope tags on the protein that is required for the antibody to be recognised, possibly explaining by ZFT is undetectable when overexpressed and exposed to boiling conditions, as has previously been observed for other transmembrane proteins (e.g. (Tsuji, 2020)).

We have clarified this in the results section, although we do not have a full explanation for this, we consider it important to share for others who may be looking at expression of these proteins.

(5) Figure 3G: It might be helpful to include an uncropped gel profile to allow readers to visualize that the main product does indeed correspond to a potential dimeric form in the native PAGE.

This has now been added in Figure S3e, thank you for this suggestion.

(6) The investigation of the impact of ZFT depletion on the apicoplast could be improved. The authors suggest that ZFT knockdown inhibits apicoplast replication based on a modest increase in elongated organelles, but the term "delayed death" is not appropriate in that case, as it is typically linked to a loss of the organelle. This is not observed here and is also illustrated by the unchanged CPN60 processing profile. So, clearly, there seems to be no strong morphological effect on the apicoplast early on after ZFT depletion. On the other hand, the authors dismiss any impact on TgPDH-E2 lipoylation (which is iron-dependent) based on the fact that the lipoylated form of the protein is still detected by Western blot. However, closer inspection of the blot in Figure 4B suggests that the intensity of the annotated TgPDH-E2 signal is reduced compared to the -ATc condition (although there might be differences in protein loading, as indicated by the control) or even with the mitochondrial 2-oxoglutarate dehydrogenase-E2, whose lipoylation is presumably iron-independent (see PMID: 16778769). This experiment should be repeated, and the results quantified properly in case something was missed, and the duration of depletion conditions perhaps extended further. Of note, it would also be worthwhile to revisit size estimations, as the displayed profiles seem inconsistent with the typical sizes of lipoylated proteins detected with the anti-lipoyl antibody (e.g., ~100 kDa for PDH-E2, ~60 kDa for branched-chain 2-oxo acid dehydrogenase, and ~40 kDa 2-oxoglutarate dehydrogenase).

We thank the reviewer for this comment. We agree that there is no strong defect on the apicoplast in the first lytic cycle and we have modified the language to remove reference to delayed death, as given the magnitude of changes associated with loss of iron and zinc, we cannot be certain about the role of the apicoplast.

Based on this suggestion, we have now quantified the levels of lipoylation of PDH-E2, BDCK-E2 and OGDH-E2 and now include this in Figure S4b, c, d. Supporting our other results, we do not see a significant change in PDH-E2 lipolyation upon ZFT knockdown. However, although OGDH-E2 lipoylation is unchanged (Figure S4c) interestingly we do see a significant increase in BDCK-E2 lipoylation (Figure S4d). This process is not expected to be directly iron related, as mitochondrial lipoylation is through scavenging rather than synthesis however, speaks to the larger mitochondrial disruption that we see. We now consider this further in the discussion.

For the sizes, we thank the reviewer for bringing this up, our apologies this was due to an error in the annotation, and we have now corrected this in the figure.

(7) In the third paragraph of the discussion, the authors mention the inability to complement ZFT loss by adding exogenous metals. One argument is the potential lack of metal access to the parasitophorous vacuole (PV). Although largely unexplored, this point could be expanded further in the discussion, as the issue of metal transport to the parasite involves not only the parasite plasma membrane but also the PV membrane. Additionally, the authors mention the absence of functional redundancy in transporters, but it would be helpful to discuss potential stage-specific or differential expression of other ZIP candidates. Transcriptomic data available on Toxodb.org could provide useful insights into this, and experimental approaches, such as RT-PCR, could be used to assess the expression of these candidates in the absence of ZFT.

On the issue of metals crossing the PV membrane, we agree that while we do not currently know mechanisms of metal transport within the infected host cell, we do have experimental confirmation that the concentration and form of the metals that we are using can impact the parasites. We show that metal treatment inhibits parasites growth (e.g. Figure 3k-n, Figure 6a-d) and we can detect the increased metals through our experiments using FerroOrange and FluroZine (Figure 7a, c). In these experiments, parasites were treated intracellularly and so we can confirm that, regardless of the mechanism, iron and zinc can reach the parasite. While entry of metals across the PV is an intriguing question, it is beyond the scope of the present work which focuses on the role of the selected transporter.

We agree that a more detailed discussion of the other ZIP transporters is warranted. We have extended this section of the discussion although for now, we cannot determine the role of the other ZIP transporters in *Toxoplasma*.

(8) In the discussion, the authors mention that « Inhibition of respiration has previously been linked to bradyzoite conversion ». To strengthen their point, the authors could mention that mitochondrial Fe-S mutants, as well as mutants affecting mitochondrial translation or the mitochondrial electron transport chain, also initiate bradyzoite conversion (PMID: 34793583). This would reinforce the connection between mitochondrial dysfunction and stage conversion.

This is an excellent point and we have added this to the discussion as follows:

“Inhibition of mitochondrial Fe-S biogenesis or mitochondrial respiration have both previously been linked to bradyzoite conversion (Pamukcu et al., 2021; Tomavo and Boothroyd, 1995), however we do not yet know the signalling factors linking iron, zinc or mitochondrial function to bradyzoite differentiation”.

(9) As a general comment on manuscript formatting, providing page and line numbers would significantly improve the manuscript's readability and allow reviewers to more easily reference specific sections. This would help address the minor issues of typos (e.g., multiple occurrences of "promotor"). I suggest a careful read-through to correct these issues.

We thank the reviewer for this comment and in the resubmitted version we have corrected these issues.

**Reviewer #2 (Recommendations for the authors):**
(1) In the alignment (Figure 1a), the BPZIP sequence is from which organism (genus, species)? It would be helpful to include this information in the figure legend.

Apologies for this oversight, this figure and section have been reworked and the species name (*Bordetella bronchiseptica)* added.

(2) In reference to Figure 1a, the authors state, "Interestingly, all parasite ZIP-domain proteins examined have a HK motif at the M2 metal binding". I was wondering if by "all" the authors mean Toxoplasma and *Plasmodium falciparum* (shown in Figure 1a) or did the authors also look at other apicomplexan parasites such as Cryptosporidium or Neospora? Is this a general feature of apicomplexan parasites?

We looked at this, and the HK motif in the M2 binding site is conserved in *Neospora Cryptosporidium*, and even the digenic gregarine *Porospora* cf. gigantea. However, in the more distantly related *Chromera* we find a HH motif at the same position. This suggests that the HK motif is present in the Apicomplexa, but not conserved in the free-living Alveolata. Although we cannot speculate on the role of this motif currently, its role in metal import in Apicomplexa does deserve future scrutiny. To reflect this finding we have modified Figure 1a and the text.

(3) In Figure 1e, to better visualize the ZFT-3HA staining at the basal pole, it would be better to omit the DAPI staining from the merged image. It is difficult to see the ZFT staining in the image of the large vacuole.

We have removed the DAPI from this image to improve clarity.

(4) Based on the "delayed-death" phenotype of the apicoplast, it is not surprising that no defects were observed in CPN60 processing or protein lipoylation. Have the authors considered measuring these phenotypes after a further round of growth (as was done for visualizing apicoplast morphology)?

We agree that changes in apicoplast function are often only seen in the second round of replication. However, here we wanted to check if ZFT depletion led to immediate changes in function of the organelle, which was not the case. It is highly likely that after the second round, we would see significant defects in the apicoplast function, however given the immediate importance of iron and zinc to many processes within the parasite, we believe that these experiments would be complicated to interpret.

(5) Depleting ZFT led to a reduction in expression levels for the mitochondrial Fe-S protein SDHB but not for a cytosolic Fe-S protein. Is it expected that less intracellular iron (via depleted ZFT) would differentially affect mitochondrial versus cytosolic Fe-S proteins?

Previous studies (e.g., Maclean et al., 2024; Renaud et al., 2025) have shown that upon direct inhibition of the cytosolic Fe-S pathway, ABCE1 is fairly stable and levels can persist for 2-3 days post treatment. However, our recent work has shown that rapid and acute depletion of iron directly (though treatment with a chelator) can lead to ABCE1 levels decreasing within 24h (Hanna et al., 2025). In the case of ZFT knockdown, due to the more gradual reduction in iron levels seen (e.g. Figure 7j) we believe the parasites are prioritising key Fe-S pathways (e.g. essential proteostasis through ABCE1), probably while remodelling metabolism (as seen in our Seahorse assays). However, there are many proteins expected to be directly impacted by iron and zinc restriction that these parasites experience, and different protein classes are expected to behave differently in these conditions.

**Reviewer #3 (Recommendations for the authors):**
(1) Is the effect on the plaque size between T7S4-ZFT (-aTc) in regular and 'high iron' conditions significant? The authors show convincingly that the plaque size is smaller due to the swapped promoter and the resulting overexpression of ZFT. But is the effect aggravated in high iron? This would be expected if excess iron were the problem.

The plaque sizes are significantly smaller in the T7S4-ZFT line under high iron compared to the untreated condition, and compared to the parental untreated line. However, if we normalise plaque size to untreated conditions for both lines, there is not a significant change in plaque size in high iron between the parental and T7S4-ZFT. This is possibly due to the concentration of iron used (200 mM), which may not be optimal to see this effect, or the time taken for plaque assays (6-7 days), which may allow the excess iron to be stored by the host cells, changing the effective concentration of parasite exposure.

(2) I struggle to understand the intracellular growth assay in Figure 5b. Here, T7S4-ZFT parasites show 25 % of vacuoles with more than 8 parasites (labelled 8+). But such large vacuoles are not observed in the parental strain. It appears as if the inducible strain grows faster even though it was earlier shown to have a fitness defect (see Figure 3j). Can you please clarify?

This is a result of rapid growth of the parental line, some vacuoles in this line lysed and initiated a new round of replication at this time point while we saw no evidence at any timepoint that ZFT-depleted parasites were able to lyse the host cell. However, the initial (24-48h post ATc addition) replication rate of the ZFT KD remains similar to the parental. In this panel, we wanted to emphasize that the major phenotype we see upon ZFT depletion is vacuole disorganisation, which we believe is linked to the start of differentiation into bradyzoites.

(3) Did the authors perform an IFA in addition to the Western blot to localize the 2nd Ty-tagged ZFT copy? It seems important to validate that the protein correctly localizes to the plasma membrane.

We have done so and now include these data in Figure S2b. Overexpression of ZFT-Ty localises to internal structures (probably vesicles) with some signal at the periphery, however, this limited expression at the periphery is sufficient to mediate the phenotypes that we see.

(4) First sentence of the abstract and introduction: The authors speak of metabolism and cellular respiration as though they are two different processes. Is respiration not part of metabolism?

This is an excellent point, we wanted to distinguish mitochondrial respiration from general cellular metabolism, but this was not clear. We have now changed this in the introduction to the below:

“Iron, and other transition metals such as zinc, manganese and copper, are essential nutrients for almost all life, playing vital roles in biological processes such as DNA replication, translation, and metabolic processes including mitochondrial respiration (Teh et al., 2024)”

(5) 2nd paragraph of the introduction: toxoplasmosis is written capitalized but should be lower case.

This has been corrected.

(6) Figure 4j legend: change 'shits parasites to a more quiescent stage' to 'shifts parasites'.

This has been corrected, our apologies.

(7) Please correct the following sentence: 'These data demonstrate ZFT depletion leads to the expression of the bradyzoite-specific markers BAG1 and DBL.' DBL is not expressed by the parasite. It is a lectin that binds to the sugars in the cyst wall.

We have now modified this in the text. The sentence now reads: “These data show that ZFT depletion leads to the expression of the bradyzoite marker BAG1 and the production of the cyst wall, as detected by DBL”*.*

(8) In the section on yeast complementation with TgZFT, the authors write: 'Based on this success, we also attempted to complement...'. Please consider changing 'Success' to something more neutral.

We have modified the text to now read: “Based on these results, we also attempted to complement”…

(9) In the discussion, the authors write: 'We see a delayed phenotype on the apicoplast, suggesting that metal import is also required in this organelle, although no apicoplast metal transporters have yet been identified.' Please consider the study *Plasmodium falciparum* ZIP1 Is a Zinc-Selective Transporter with Stage-Dependent Targeting to the Apicoplast and Plasma Membrane in Erythrocytic Parasites (PMID: 38163252).

We thank the reviewer for the note and have modified the text to include this and the reference. Please see below:

“Iron is known to be required in the apicoplast (Renaud et al., 2022), zinc also may be required, as the fitness-conferring Plasmodium zinc transporter ZIP1 is transiently localised to the apicoplast (Shrivastava et al., 2024), although the functional relevance of this localisation has not yet been established”.

(10) The authors write: 'Iron is known to be required in the apicoplast (Renaud et al., 2022), although a potential role for zinc in this organelle has not yet been established.' The role for zinc in the apicoplast may not have been shown formally, but surely among its hundreds of proteins, and those involved in replication and transcription, there are some that depend on zinc...?

Yes, we agree it would make sense, however multiple searches using ToxoDB and the datasets from Chen et al (2025) were unable to find any apicoplast-localised proteins with zinc-binding domains. We cannot exclude that zinc is in the apicoplast, and the results from *Plasmodium* (Shrivastava et al., 2024) may suggest that is, however currently we do not have any evidence for its role within this organelle.

References

DiDonato, R.J., Roberts, L.A., Sanderson, T., Eisley, R.B., Walker, E.L., 2004. Arabidopsis Yellow Stripe-Like2 (YSL2): a metal-regulated gene encoding a plasma membrane transporter of nicotianamine-metal complexes. Plant J 39, 403–414. https://doi.org/10.1111/j.1365-313X.2004.02128.x

Hanna, J.C., Shikha, S., Sloan, M.A., Harding, C.R., 2025. Global translational and metabolic remodelling during iron deprivation in *Toxoplasma gondii*. https://doi.org/10.1101/2025.08.11.669662

Maclean, A.E., Sloan, M.A., Renaud, E.A., Argyle, B.E., Lewis, W.H., Ovciarikova, J., Demolombe, V., Waller, R.F., Besteiro, S., Sheiner, L., 2024. The *Toxoplasma gondii* mitochondrial transporter ABCB7L is essential for the biogenesis of cytosolic and nuclear iron-sulfur cluster proteins and cytosolic translation. mBio 15, e00872-24. https://doi.org/10.1128/mbio.00872-24

Pamukcu, S., Cerutti, A., Bordat, Y., Hem, S., Rofidal, V., Besteiro, S., 2021. Differential contribution of two organelles of endosymbiotic origin to iron-sulfur cluster synthesis and overall fitness in Toxoplasma. PLoS Pathog 17, e1010096. https://doi.org/10.1371/journal.ppat.1010096

Renaud, E.A., Maupin, A.J.M., Berry, L., Bals, J., Bordat, Y., Demolombe, V., Rofidal, V., Vignols, F., Besteiro, S., 2025. The HCF101 protein is an important component of the cytosolic iron–sulfur synthesis pathway in *Toxoplasma gondii*. PLoS Biol 23, e3003028. https://doi.org/10.1371/journal.pbio.3003028

Shrivastava, D., Jha, A., Kabrambam, R., Vishwakarma, J., Mitra, K., Ramachandran, R., Habib, S., 2024. *Plasmodium falciparum* ZIP1 Is a Zinc-Selective Transporter with Stage-Dependent Targeting to the Apicoplast and Plasma Membrane in Erythrocytic Parasites. ACS Infect. Dis. 10, 155–169. https://doi.org/10.1021/acsinfecdis.3c00426

Teh, M.R., Armitage, A.E., Drakesmith, H., 2024. Why cells need iron: a compendium of iron utilisation. Trends in Endocrinology & Metabolism 35, 1026–1049. https://doi.org/10.1016/j.tem.2024.04.015 Tomavo, S., Boothroyd, J.C., 1995. Interconnection between organellar functions, development and drug resistance in the protozoan parasite, *Toxoplasma gondii*. International Journal for Parasitology 25, 1293–1299. https://doi.org/10.1016/0020-7519(95)00066-B.